



Solid Earth

# Biotite supports long-range diffusive transport in dissolution–precipitation creep in halite through small porosity fluctuations

**Berit Schwichtenberg**[1]**, Florian Fusseis**[1]**, Ian B. Butler**[1]**, and Edward Andò**[2]

[1]The University of Edinburgh, School of Geosciences, Edinburgh, UK
[2]Université des Alpes, CRNS, Laboratoire 3SR, Grenoble, France

**Correspondence:** Berit Schwichtenberg (berit.schwichtenberg@outlook.com)

**Abstract.** Phyllosilicates are generally regarded to have a reinforcing effect on chemical compaction by dissolution–precipitation creep (DPC) and thereby influence the evolution of hydraulic rock properties relevant to groundwater resources and geological repositories as well as fossil fuel reservoirs. We conducted oedometric compaction experiments on layered NaCl–biotite samples to test this assumption. In particular, we aim to analyse slow chemical compaction processes in the presence of biotite on the grain scale and determine the effects of chemical and mechanical feedbacks. We used time-resolved (4-D) microtomographic data to capture the dynamic evolution of the porosity in layered NaCl–NaCl/biotite samples over 1619 and 1932 h of compaction. Percolation analysis in combination with advanced digital volume correlation techniques showed that biotite grains influence the dynamic evolution of porosity in the sample by promoting a reduction of porosity in their vicinity. However, the lack of preferential strain localisation around phyllosilicates and a homogeneous distribution of axial shortening across the sample suggests that the porosity reduction is not achieved by pore collapse but by the precipitation of NaCl sourced from outside the NaCl–biotite layer. Our observations invite a renewed discussion of the effect of phyllosilicates on DPC, with a particular emphasis on the length scales of the processes involved. We propose that, in our experiments, the diffusive transport processes invoked in classical theoretical models of DPC are complemented by chemo-mechanical feedbacks that arise on longer length scales. These feedbacks drive NaCl diffusion from the marginal pure NaCl layers into the central NaCl–biotite mix-

ture over distances of several hundred micrometres CE1 and several grain diameters. Such a mechanism was first postulated by Merino et al. (1983).

## 1 Introduction

Chemically controlled compaction influences groundwater resources, geological waste repositories and $CO_2$ sequestration as well as fossil fuel reservoirs. One of the major processes involved in chemical compaction in the Earth's upper crust is dissolution–precipitation creep (DPC), a diagenetic and low-grade metamorphic deformation process that significantly contributes to cementation and the reduction of porosity in sedimentary rocks (Rutter, 1983; Green, 1984; Tada and Siever, 1989; Gratier et al., 2013). Due to its impact on the diagenetic evolution of sediments, it is crucial to study how DPC contributes to the dynamic change of hydraulic rock properties during compaction.

DPC describes a sequential chemo-mechanical process in a non-equilibrium system consisting of a solid phase and its associated fluid under non-hydrostatic pressure conditions (Rutter, 1983; Gratier et al., 2013). The three successive steps in the sequence are (i) dissolution of material at stressed grain contacts, (ii) diffusive mass transport through an intergranular fluid and (iii) local reprecipitation of dissolved material at low-stress sites (e.g. open pores, veins) (Rutter, 1983; Tada and Siever, 1989; Gratier et al., 2013).

Phyllosilicates have been recognised to have a reinforcing effect on the dissolution process (e.g. Heald, 1956; Weyl,

1959; Gratier, 1987) and act as loci for enhanced DPC. Whether this is due to enhanced reaction kinetics or effective transport pathways is still under debate (Gratier, 1987) and may depend on the rate-controlling process. Macente et al. (2018) explored this effect using time-resolved X-ray microtomography to document chemical compaction in NaCl–biotite mixtures. They found that the increased porosity loss in the biotite-bearing layer did not lead to an increased strain localisation. These observations pointed towards infilling of porosity with material sourced outside the biotite-bearing layer rather than pore collapse, suggesting long-scale diffusive transport of dissolved NaCl.

Reviewing diffusive transport during DPC shows that in theory four pathways for material transport need to be considered. On the one hand, there is intragranular Nabarro–Herring (Herring, 1950) and Coble creep (Coble, 1963), and on the other hand, diffusion through a free fluid either within grain boundaries or the open pore space (Durney, 1976; McClay, 1977). While the intragranular diffusion mechanisms are activated at elevated temperatures (Raj, 1982), it is commonly accepted that diffusive transport during low-temperature DPC occurs through an intergranular fluid film along grain boundaries (e.g. Raj, 1982; Rutter, 1983; Gratier, 1987). Herein, transport distances do not exceed the grain size and dissolution sites are locally connected to precipitation sites (Raj, 1982; Gratier, 1987; Groshong Jr., 1988; Croize et al., 2013; Gratier et al., 2013). However, theoretical approaches (Durney, 1972, 1976; Merino et al., 1983; Lehner, 1995; Gundersen et al., 2002) as well as field evidence (Mimran, 1977; Buxton and Sibley, 1981) challenge this interpretation and suggest that long-distance diffusive transport in the open pore space needs to be considered as well.

Very few experimental data document the transport length scales of DPC. In this contribution, we build upon the work of Macente et al. (2018), and report the outcomes of further experiments on analogue samples of NaCl, using an improved experimental setup and advanced analysis protocols for our time-resolved X-ray microtomography data. In contrast to Macente et al. (2018), who emphasise the impact of phyllosilicates upon the evolving porosity, our aim was to qualitatively determine length scales of diffusive transport in a dynamically evolving porosity during DPC and compare the results to existing transport models. The results of our experiments show that the diffusive transport length scales of DPC may exceed the grain scale without the contribution of advective transport, a phenomenon that has not been observed experimentally yet.

## 2  Materials and methods

### 2.1  Introduction

Aggregates containing sodium chloride (NaCl) and biotite were used to analyse the effect of DPC in granular materials. As previous studies have shown (e.g. Spiers et al., 1990; Peach, 1991; Bons and Urai, 1994; Macente, 2017; Macente et al., 2018), analogue materials are suitable to study deformation mechanisms at moderate $P - T$ conditions in experiments of a tractable duration, and the results can be extrapolated to natural conditions. We chose NaCl as its solubility at low pressures and temperatures is high compared to other constituent minerals of sedimentary rocks (Trurnit, 1968) and it has previously been used for compaction experiments at room temperature (Spiers and Schutjens, 1990; Schutjens and Spiers, 1999; Renard et al., 2001, 2004; Gratier et al., 2013). It is further considered as a host rock for geological nuclear waste repositories (e.g. Hansen and Leigh, 2011; von Berlepsch and Haverkamp, 2016), and its deformation behaviour is well characterised (Carter and Hansen, 1983; Urai et al., 2008).

### 2.2  Sample preparation

Two layered samples and one homogeneous sample were prepared for oedometric compaction experiments in low X-ray attenuation oedometer cells (Fig. 1). A detailed description of the cell design can be found in Macente (2017). The first sample (SBS) contained a central NaCl–biotite layer and two adjoining layers of pure NaCl as well as two layers of glass beads. The latter maintain permeability at the sample ends. For the second sample (SB), we increased the ratio of the NaCl–biotite layer and removed the top NaCl layer. A third pure NaCl sample (S) served as a reference.

Masses for all components were calculated to meet the dimensions of cylindrical samples with 5 mm diameter and a desired starting height of 8 mm. For the unconsolidated samples, an initial porosity of $\sim 40\,\%$ was assumed which was also included into the calculation. The absolute heights of the samples were determined after loading the oedometers with the granular aggregate. Analytical-grade NaCl and natural biotite were chosen as main components for the samples, as the difference in the X-ray attenuation results in a sufficient contrast in the reconstructed $\mu$CT data. The granular NaCl was sieved to a grain size of 250–300 $\mu$m. Biotite of granodioritic origin (Lone Grove pluton, Texas, e.g. Zartman, 1964) was pre-processed by mineral separation techniques (conducted at GFZ Potsdam, who supplied the biotite mineral separate) and sieved to a grain size of 200–500 $\mu$m (maximum grain diameter). Acid-washed glass beads with a diameter of 212–300 $\mu$m were added as chemically inert and permeable top and base layers.

The individual sample components were introduced into the sample cell by wet loading. Saturated NaCl brine was in-

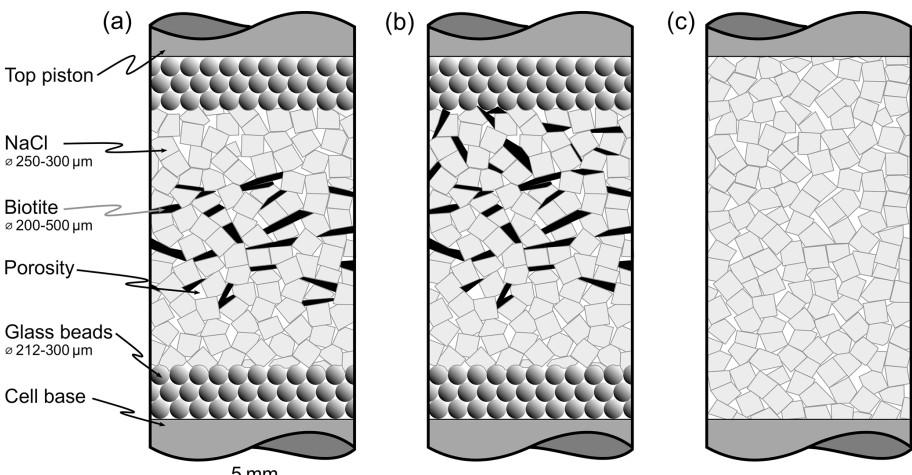

**Figure 1.** Schematic sketches of sample configurations of **(a)** the NaCl–biotite–NaCl sample (SBS), **(b)** the NaCl–biotite sample (SB) and **(c)** the pure NaCl reference sample (S) before deformation. Different shades of grey depict the single components but are not related to their appearance in the tomography scans. White angular patches represent the brine saturated pore space with arbitrary distribution of shape and size, light grey angular to square objects are cubic grains of analytical-grade NaCl with a sieved grain size of 250–300 μm, and black elongated angular shapes are biotite grains of 200–500 μm size. At the top and bottom of the samples in panels **(a)** and **(b)**, dark grey circles describe acid-washed glass beads of 212–300 μm diameter, which were inserted as a chemically inert layer. The original samples have a diameter of 5 mm; note that the sketches are not to scale.

jected into the bottom of the sample cell. For the two biotite-bearing samples (SBS and SB) glass beads and NaCl were sequentially poured into the brine followed by a homogeneous slurry of NaCl and 20 wt% biotite. For the pure NaCl sample, dry NaCl (not a slurry) was poured into the injected brine. All samples were covered at the top and bottom with a disc of filter paper to prevent blockage of the fluid inlet and outlet.

After the piston was twisted into the oedometer, the samples were flushed with brine to saturate the entire pore space with fluid. At this stage, a low axial load was applied to the top piston which kept the piston in place but was balanced by the pore pressure so that the effective load remained zero.

## 2.3 Experimental setup

The experimental setup was designed to run three oedometric compaction experiments simultaneously (Fig. 2). We adopted the basic setup from Macente et al. (2018) and added a thermally insulated environment, vibration damping, pressurised fluid pumps and chemically inert glass bead layers to gain better control of parameters and establish a better characterised system. The oedometer cells as described in Macente et al. (2018) were modified by sealing the sample cell with an O ring around the piston in order to withstand fluid pressure of 0.5 MPa. The latter was applied in two different ways. For the SBS sample, we used Cetoni neMESYS syringe pumps feeding a fluid transfer vessel that isolated the metal-corrosive brine. The transfer vessel was composed of a silicone tube filled with saturated NaCl brine inside a pressure resistant glass column. Saturation of the brine was guaranteed by the presence of solid NaCl in the reservoir. For the pure NaCl and the SB samples, we used a plastic syringe that contained the corrosive brine itself and was driven by a pneumatic actuator. The experimental setup allowed maintaining a moderate pore fluid pressure to suppress gas bubbles while suppressing fluid advection.

The axial load was also applied by gas-pressure-driven pneumatic actuators when the oedometer cells were placed in loading frames following the construction of Macente (2017).

The experiments were run inside a thermally insulated box where the temperature was logged and found to be stable within ± 1.7 °C over the course of the experiments. For the acquisition of microtomographic data, the oedometers were disconnected from the fluid and pneumatic capillaries and mounted into the microtomography scanner (see next section).

Upon initial loading, the unconsolidated granular samples were held at a constant effective load of 6.64, 6.77 and 10.5 MPa for samples S, SBS and SB, respectively, for 60 min. After an initial compaction of 9 %–18 %, a reference scan was acquired of the starting aggregate.

The loading parameters were maintained throughout the experiments and consistently monitored over a total duration between 1089 and 1932 h. Following Macente (2017) and Macente et al. (2018), the conditions for the first experimental suite (SBS) were chosen to be similar which allowed comparison of the data with each other. For the second experimental suite (SB and S), the effective load was increased

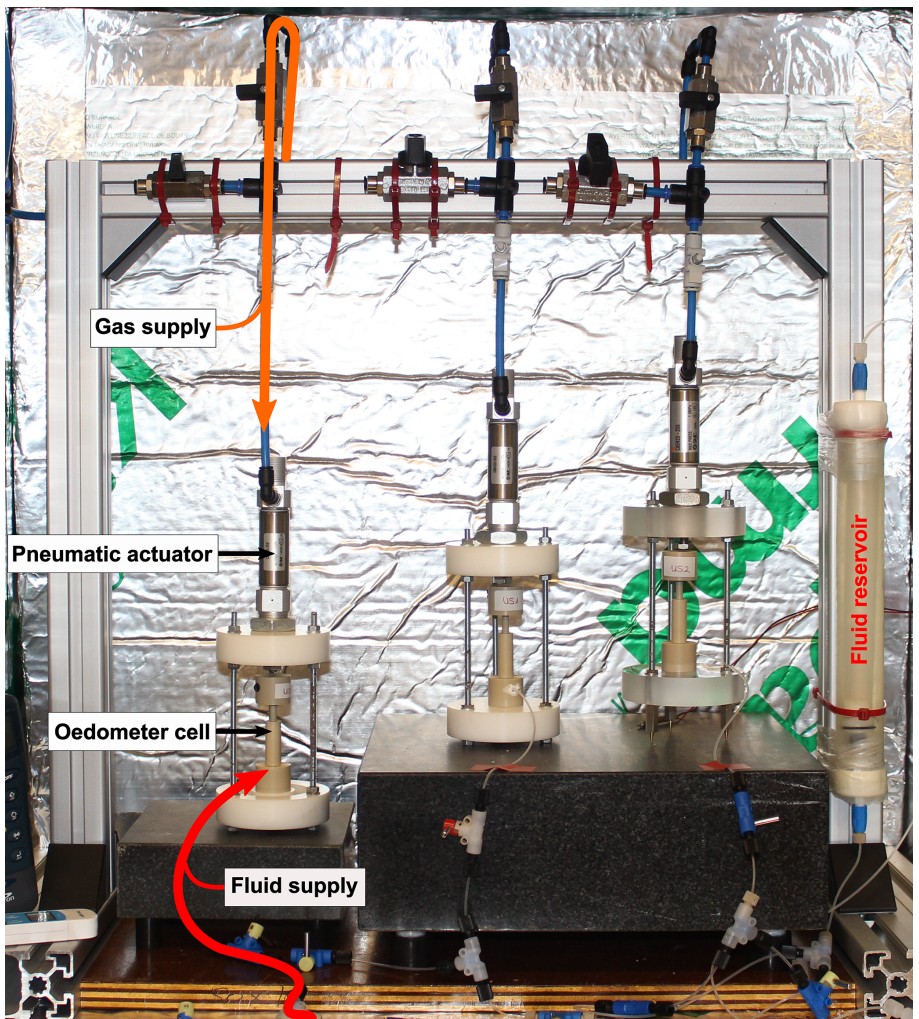

**Figure 2.** Experimental setup for oedometric compaction experiments conducted at the University of Edinburgh. Three samples could be loaded simultaneously. Oedometer cells contained the cylindrical samples and were confined in straining frames during deformation. The load was applied by gas-driven pneumatic actuators installed at the top of each frame. Cetoni neMESYS high-pressure syringe pumps were used to supply saturated NaCl brine via a fluid reservoir in order to maintain a pore fluid pressure sufficient to suppress gas bubbles in the samples. For the second experimental suit we replaced the fluid reservoir and high-pressure pump with a brine-filled syringe that was driven by another pneumatic actuator (not shown here). The oedometer cells were placed on acoustically dampened gabbro blocks to avoid external vibrations to reach the cells.

in order to increase the strain rate according to the rate law for diffusion-controlled DPC (Spiers et al., 2004).

## 2.4   Data acquisition

During the experiment, the samples were scanned ex situ on the X-ray microtomography instrument at the School of Geosciences, University of Edinburgh, in regular intervals (for acquisition parameters, see Table A2 in the Appendix). To enable this, the oedometers were unloaded and temporarily removed from the loading frames. In total, 19 scans (SBS sample), 10 scans (SB sample) and 5 scans (S sample) were acquired over a total duration of 1619, 1932 and 1089 h, respectively. At the beginning, scans were collected in shorter time intervals to image the rapidly progressing deformation within the first 200 h of the experiment. As the compaction slowed down, the intervals between each scan were gradually extended to monitor chemical compaction processes. The time-resolved 3-D data series obtained in this way were then combined into three 4-D data sets capturing the dynamic evolution of the porosity in the different samples.

## 2.5   Data processing

After each scan, Octopus® software v. 8.9 (Dierick et al., 2004) was used to reconstruct the μCT data from the radiographic projections. The resulting stack of 2-D images, which contains a virtual representation of half of the sample,

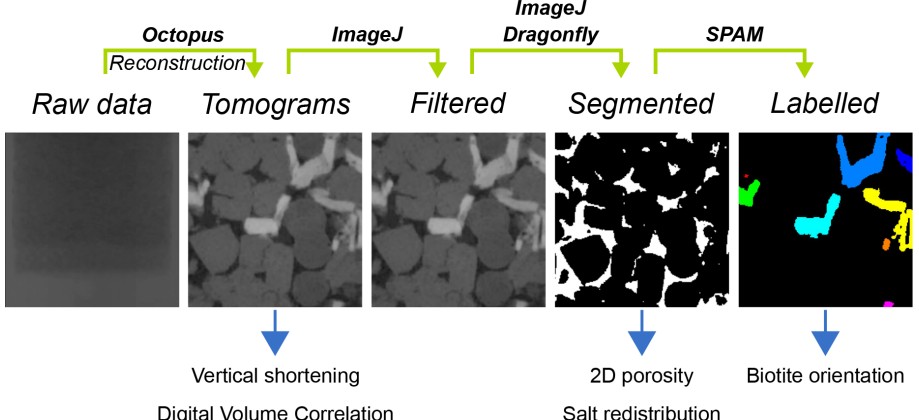

**Figure 3.** Workflow used for data processing. At the top, the specific software used in each processing step is given, while the information gained from the individual steps is listed below a representative 2-D slice of the particular processing result. The individual steps are based on each other going from the left (the unprocessed radiograph) to the right (highly processed data).

covers a volume of $5 \times 10^3\,\mu m^3$. Each time step comprises two scans, vertically translated to cover the entire sample including the sample base and the top piston. Reconstructed image stacks were merged using the image processing software Avizo® v. 9.2. The relative shortening of the sample was calculated using

$$\varepsilon_t = \frac{L_0 - L_t}{L_0}, \tag{1}$$

with the relative vertical shortening $\varepsilon_t$ at the time $t$, the initial sample height $L_0$ and the sample height $L_t$ at the time $t$. The height of each sample was determined from the merged $\mu$CT scans. The $\mu$CT data were prepared for analysis by pre-processing in ImageJ (Schindelin et al., 2012). An uneven background of the reconstructed images was adjusted using the background correction plugin BaSiC (Peng et al., 2017) with the regularisation parameter $\lambda_D = 0.69$ and $\lambda_F = 1.71$ for the estimated dark-field and flat-field images, respectively. Denoising of the data was conducted by removing bright outliers (threshold of 50) of 1.5 px and using the non-local means filter (Buades et al., 2011) with a standard deviation of $\sigma = 3$ and a smoothing factor of 1. Biotite grains and porosity were segmented as single classes from the images, applying the machine learning tool "Trainable Weka Segmentation" (Arganda-Carreras et al., 2017). The latter was also used to segment NaCl grains in the pure NaCl sample but not in the two biotite-bearing samples. Due to a lack of contrast between the grey scales of NaCl, glass beads and the outer rims of biotite grains, it was not possible to use simple segmentation techniques. Instead we applied the "Deep Learning Segmentation" of Dragonfly software, version 2020.2 (Object Research Systems , ORS) to discretely segment the NaCl. The resulting binarised image stacks were used for image analysis to quantify the porosity evolution and NaCl migration.

### 2.5.1 Porosity measurements

The evolving porosity was determined during compaction as two-dimensional porosity which was measured along the direction of the loading axis as the relative area of the two-dimensional binarised images. The results were plotted as the porosity on the abscissa and the number of slices along the loading axis on the ordinate (Fig. 13). Errors for this analysis were determined using the probability measured in the Trainable Weka Segmentation. The absolute error of the porosity measurement was defined as the difference between the number of pixels belonging to the porosity with a certainty of $\geq 90\,\%$ to the total number of porosity pixels measured from the segmented data. The absolute error is given in percent, as it refers to the porosity which itself is a relative number.

### 2.5.2 NaCl redistribution measurements

In order to quantify the amount of NaCl migrating within the sample, we used two different methods to isolate a pure NaCl signal from the rest of the sample. The first method is similar to the measurement of the 2-D porosity. For each slice along the loading axis of a microtomography scan, we discretely segmented the NaCl in the grey-scale image using Trainable Weka Segmentation for the pure NaCl sample and the Deep Learning Segmentation of Dragonfly software for both biotite-bearing samples. From the segmented data, we measured the relative area of NaCl in the two-dimensional image and calculated its proportion relative to the cross section of the sample. The results were plotted as relative NaCl content on the abscissa and the number of slices along the loading axis on the ordinate (Fig. 14).

Our second approach towards quantifying the NaCl redistribution in the sample was based on 3-D volumetric measurements of segmented NaCl. The volumes of NaCl, biotite and porosity were measured relative to a subvolume in the

**Table 1.** Correlation parameters used to conduct digital volume correlation for different data series. The correlation window size, usually used in digital image correlation (DIC) applications, corresponds to $1 + 2\times$ the half window size given in the table. The threshold value is simply implemented to skip correlation windows placed outside the sample based on their mean grey value.

| DVC parameters | SBS | SB | S |
|---|---|---|---|
| Correlation half window size | 30 | 30 | 30 |
| Measurement point spacing | 15 | 15 | 15 |
| Low-grey-level threshold | 4000 | 5000 | 16 000 |

compacting biotite-bearing layer (for location of the subvolume, see Fig. A1 in the Appendix). We selected the subvolumes at fixed locations within the biotite-bearing domains of both samples that mimic the compaction of the respective layer. The extent of the subvolume parallel to the loading axis was defined by two prominent biotite grains at the top and bottom of each biotite-bearing layer which were easy to identify with progressing compaction of the sample. Perpendicular to that dimension, the base of the subvolume was chosen as a 500 px × 500 px◼TS1 square in the centre, which is representative of the sample but excludes the contact area of the sample to the cell. This approach allowed us to measure the evolution of the NaCl volume within the biotite-bearing layer with progressing deformation, with biotite acting as an internal standard. The results were plotted as NaCl volumes relative to a decreasing subvolume in the compacting biotite-bearing layer (Fig. 15).

## 2.6 Digital volume correlation

Digital volume correlation (DVC) was used to quantify deformation on the grain scale between pairs of consecutive 3-D $\mu$CT data sets using the Python packages SPAM (Stamati et al., 2020) and TomoWarp2 (Tudisco et al., 2017). While SPAM calculates the displacement field between two data sets, TomoWarp2 uses the latter as an input to calculate the strain field. In a first step, a grid of regularly spaced measurement points was defined in the reference image. Each of these points is the centre point of a cubic correlation window. A corresponding cube in the deformed image was transformed with a homogeneous deformation function in order to minimise a sum-of-squared-differences error function. The correlation parameters were individually tuned for each data series and are documented in Table 1. The DVC analysis was completed by strain determination. A finite strain tensor $U$ and its first two invariants – the volumetric strain (Eq. 3) and the deviatoric strain (Eq. 4) – were calculated from the locally measured displacement component of the deformation function.

$$U = U_{\text{isotropic}} \cdot U_{\text{deviatoric}} \tag{2}$$

While the isotropic component of the strain tensor is equal to the volumetric strain and describes a change in volume, the deviatoric strain describes the deformation at constant volume.

$$U_{\text{isotropic}} = J^{1/3} \cdot \mathbf{I} \tag{3}$$

$$U_{\text{deviatoric}} = \frac{1}{J^{1/3}} \cdot U, \tag{4}$$

where $U_{\text{isotropic}}$ is the isotropic or volumetric strain, and $U_{\text{deviatoric}}$ the deviatoric strain. $\mathbf{I}$ is the identity matrix, $J$ the determinant of the strain tensor, and its exponent refers to the dimension of the problem which is three-dimensional in this case. It is important to note that the given definition of strain applies to strain on the grain scale only and needs to be differentiated from the 1-D macro-strain that reflects the vertical shortening and compaction of the bulk samples.

The mean strain rates were obtained by dividing the magnitude of the strain tensor by the time (in seconds) between two analysed data sets.

With label analysis, also implemented in SPAM, we determined the rotation and rearrangement of single biotite grains with progressing deformation based on labelled and binarised data sets. For each microtomography scan, we segmented the biotite grains as described in Sect. 2.5 and applied a watershed algorithm to separate the grains into individual particles with an allocated label. Further, we calculated the eigenvalues and eigenvectors from each particle's moment of inertia, which is in the case of biotite directly related to the shape and orientation of the particle. The orientations are represented as maximum eigenvectors perpendicular to the basal planes of the grains and plotted as densities in Lambert projections with the vertical loading axis in the centre of the plot (Fig. 7).

## 3 Results

### 3.1 Bulk compaction behaviour

#### 3.1.1 Vertical shortening

The bulk compaction of the samples was monitored as vertical shortening and compaction rate over a total duration of 1089 h, 1619 h and 1932 h, for the reference sample (S), the SBS and the SB samples, respectively.

All three samples showed a non-linear decrease in height. The SBS sample accommodated a total strain of $\sim 25\%$ over 1619 h of compaction. Initially, the sample shortened by $\sim 9\%$ within the first 16 h of compaction, resulting in a steep gradient of the compaction curve, while in the following interval the compaction rate gradually decreased and stabilised after about 280 h, indicating apparent steady-state deformation (Fig. 4). The compaction rates of the SB sample and the pure NaCl reference sample followed similar trends. The SB sample accommodated a total vertical shortening of $\sim 26\%$

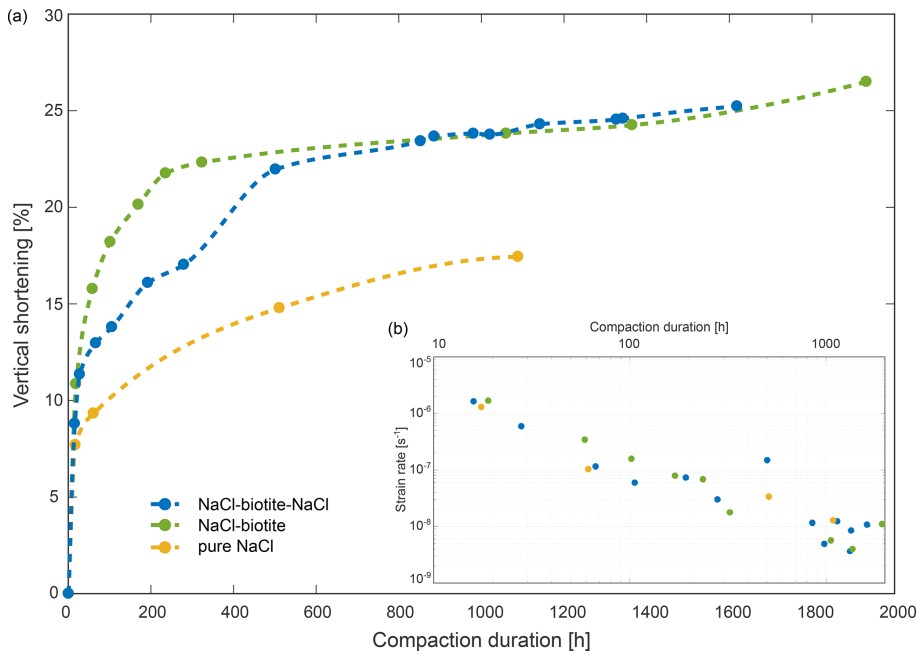

**Figure 4.** Bulk compaction curves **(a)** and strain rates **(b)** derived from $\mu$CT data for the three samples, SBS in blue, SB in green and pure NaCl in yellow. Panel **(a)** shows the vertical shortening of the samples with progressing deformation. We interpret the non-linearity of the curves as the transition from a loosely packed aggregate where mechanical compaction significantly contributes to the strain rate, to an interlocked aggregate dominated by chemical compaction. Note that all three samples show a similar trend with apparent steady-state deformation established after $\sim 250$ h. Data points were connected by spline interpolation. In panel **(b)**, we plotted the corresponding bulk strain rates over the compaction duration.

after 1932 h of which $\sim 11\%$ occurred within the first 19 h of deformation. This sample reached apparent steady-state deformation after about 324 h.

As the compaction of the reference sample was stopped after 1089 h, the total shortening of this sample was with $\sim 17\%$, lower than the total shortening of the biotite-bearing samples. At comparable compaction stages of 1020 and 1060 h, for the SBS and SB sample, respectively, the difference between the reference and the biotite-bearing samples was approximately $\sim 10\%$ (Fig. 4a). It is notable that the final scans in all three experiments were acquired before the compaction ceased.

Analysis of the bulk strain rates ($\dot{\varepsilon}$) with progressing compaction support these findings. On a double logarithmic scale (Fig. 4b), strain rates for all three samples decreased approximately linearly and dropped in total by 2 orders of magnitude from $\sim 10^{-6}$ to $\sim 10^{-8}$ s$^{-1}$. Within the first 200 h of compaction, the strain rates decreased by 1 order of magnitude. The apparent steady-state deformation interval was characterised by a constant strain rate gradient of $\sim -5 \times {}^{-11}$ s$^{-1}$. Linear regressions for the three strain rates over time ($t$) have

the form of

$$\dot{\varepsilon} = t^{-1.09} \times e^{-10.80} \tag{5}$$

$$\dot{\varepsilon} = t^{-1.26} \times e^{-9.81} \tag{6}$$

$$\dot{\varepsilon} = t^{-1.01} \times e^{-11.16}, \tag{7}$$

for the SBS, SB and pure NaCl sample, respectively.

### 3.1.2 Microstructures and compaction accommodation

Vertical slices through the geometrical centre of $\mu$CT data illustrate the evolution of the microstructure (Fig. 5). In the biotite-bearing samples' porosity reduction, a change of the cubic habit of the NaCl grains and the establishment of flat interphase boundaries between NaCl and biotite grains can be observed over 100 and 60 h for the SBS and the SB sample, respectively. Within the biotite-bearing layers, characteristic evidence for DPC (dissolution leading to grain indentation and reprecipitation of dissolved matter) was enhanced along phase boundaries (Figs. 5a–h and 6a–j). However, the pure NaCl layers as well as the NaCl reference sample showed, to a smaller extent, similar dissolution structures at later compaction stages (Fig. 6k–n). The efficiency of the process becomes obvious on the grain scale (Fig. 6a–j and animation S1 in the Supplement, TS2 which shows a single NaCl grain enclosed by two biotite grains with progressing

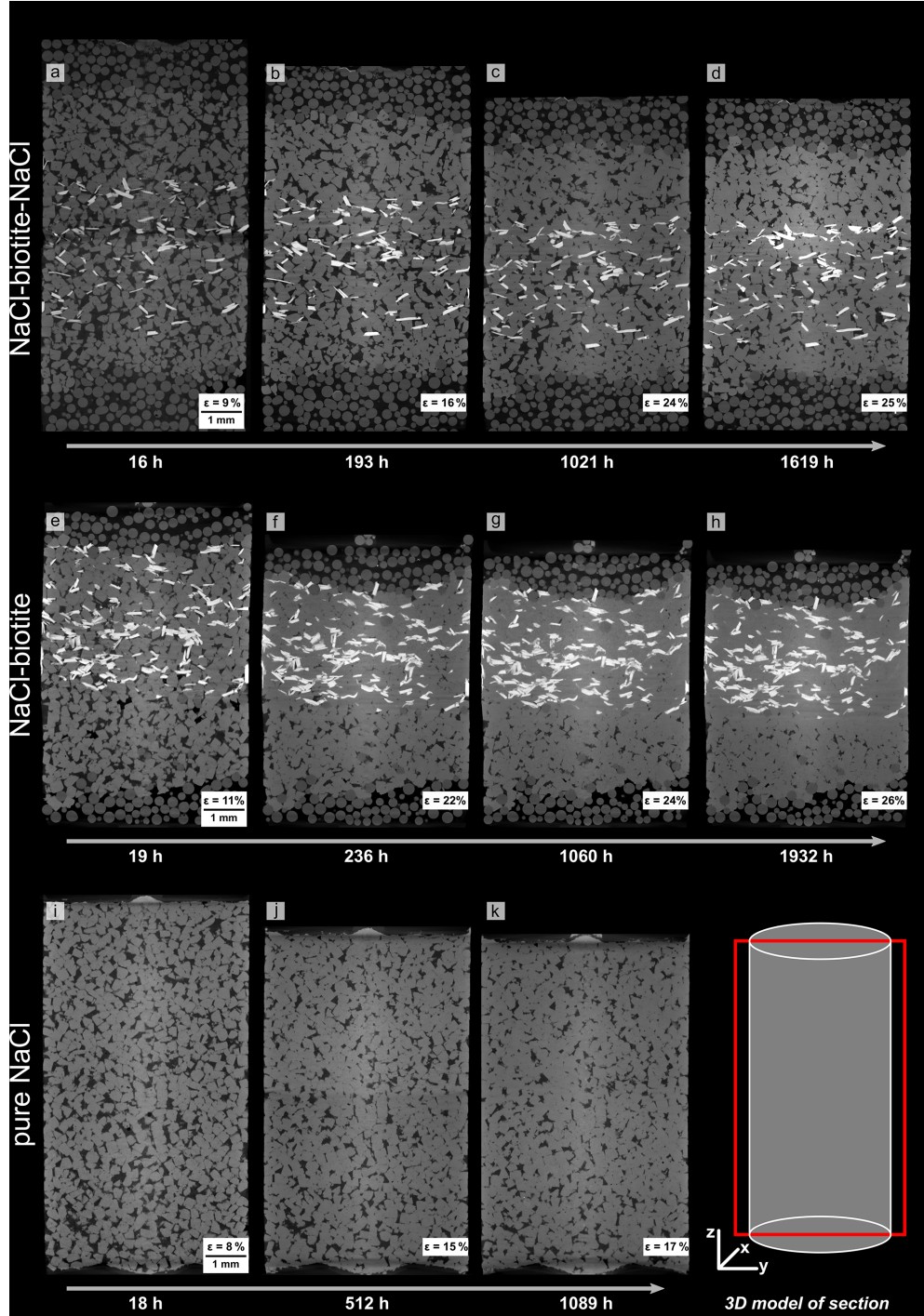

**Figure 5.** Vertical slices CE2 through absorption $\mu$CT scans at different stages of compaction. The 3-D model in the lower right corner shows the location of the section within the bulk samples. Different shades of grey refer to different phases present in the samples (in black: brine-filled pore space, dark grey: glass beads, grey: NaCl grains and light grey: biotite). The top and middle rows show the NaCl–biotite samples SBS **(a–d)** and SB **(e–h)**, respectively, the bottom row displays the pure NaCl sample **(i–k)**. Panels **(b)**, **(f)** and **(j)** show first signs of porosity reduction and indentation of NaCl grains which we interpret as indicators for active dissolution–precipitation creep and which is continuing throughout the experiment. Note that the final scan of the SB sample **(h)** shows no remaining porosity in the biotite-bearing layers, whereas it is still clearly visible in the SBS **(d)** and the pure NaCl sample **(k)**.

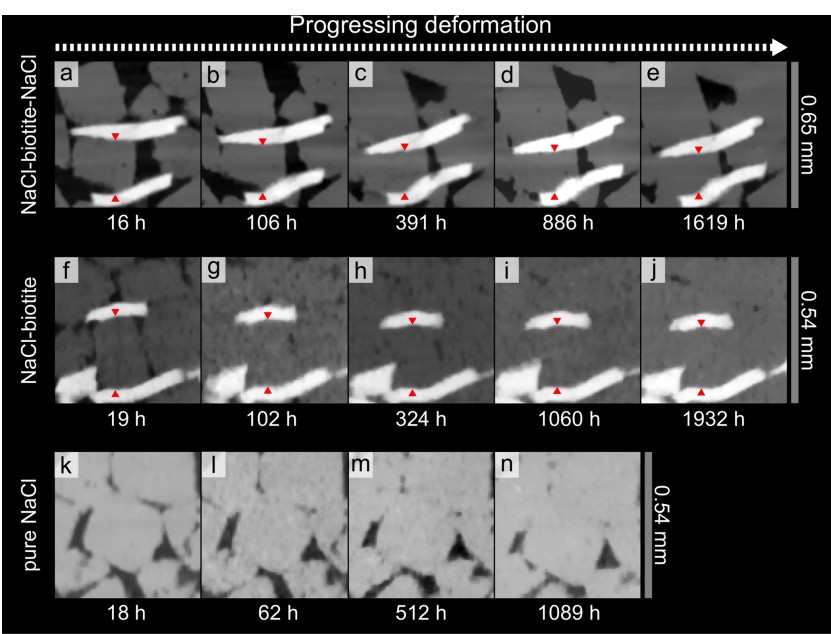

**Figure 6.** Sequence of time steps showing active dissolution–precipitation creep at the interphase boundaries of a NaCl grains with two biotite grains in the SBS **(a–e)** and SB samples **(f–j)**. While the sizes of the NaCl grains are reduced without showing any signs of brittle deformation, interphase contacts with the mica become flattened, indicating dissolution of NaCl. Comparison of panels **(a)** and **(e)** and **(f)** and **(j)** shows that the vertical strain accommodated by the NaCl grains is $\sim 26\%$ in the SBS sample and $\sim 22\%$ in the SB sample. Note that the biotite grains have experienced only little rotation during deformation, although 1603 and 1913 h of compaction are in between the first and the last images. In the SBS sample, the rotation of the top biotite grain occurs prior to the apparent steady state. Red arrows are used as markers for strain calculations. The pure NaCl sample **(k–n)** on the other hand shows less activity of DPC. However, here as well contacts between grains are flattened and pore space reduced.

deformation). We found that the orientation of the enclosing biotite grains did not change significantly as the volume of the NaCl grains was reduced. We calculated the vertical shortening of the single NaCl grain to be $\sim 26\%$ in the SBS sample and $\sim 22\%$ in the SB sample, which is similar to the shortening calculated for the bulk biotite-bearing samples (see Sect. 3.1.1).

The orientation of the maximum eigenvector of individual biotite grains was plotted in Lambert projections (Fig. 7; see the Appendix for an explanation on how to read a Lambert projection). For both biotite-bearing samples, areas with a high density of data points were located close to the centre of the plots, revealing that the maximum eigenvectors of single biotite grains were oriented vertically and did not significantly rotate away from the vertical loading axis after an instantaneous mechanical rearrangement of the grains.

Our analysis showed that the compaction was accommodated throughout the samples, independent of their composition. Only the glass bead layers did not accommodate further compaction after an initial mechanical rearrangement. Figure 8 shows the vertical displacement rate of the biotite-bearing layer and the bulk sample for different increments of progressing deformation for the two biotite-bearing samples. The time intervals which correspond to the individual increments are listed in the table below the plot. At the be-

ginning of the experiment, the rate of both bulk samples was elevated compared to the biotite-bearing layers. However, as compaction approached an apparent steady state, the vertical displacement rates of the biotite-bearing layer and the bulk sample became comparable and both asymptotically approached a value of zero.

## 3.2 Strain analysis

DVC was used to analyse a locally resolved strain field and associated strain rates. Figures 9–11 show the deviatoric strain and volumetric strain for the three samples at different stages during the deformation. Deviatoric strain maxima in the two biotite-bearing samples (Figs. 9 and 10) were located within the biotite-bearing layers but not exclusively. In both samples, pure NaCl domains adjacent to the biotite-bearing layer were also affected by higher strains. Note that the activity in the top NaCl layer of the SBS sample seems to be lower than in the bottom NaCl layer of the same sample. In addition to that, the deviatoric strain maxima in the SBS sample correlate with negative volumetric strain (compaction). A similar correlation was found in the SB sample. Here as well, no layer-specific strain pattern with strain maxima located in the biotite-bearing layer emerged. Further, the strain distribution in the biotite-bearing samples was comparable to the one ob-

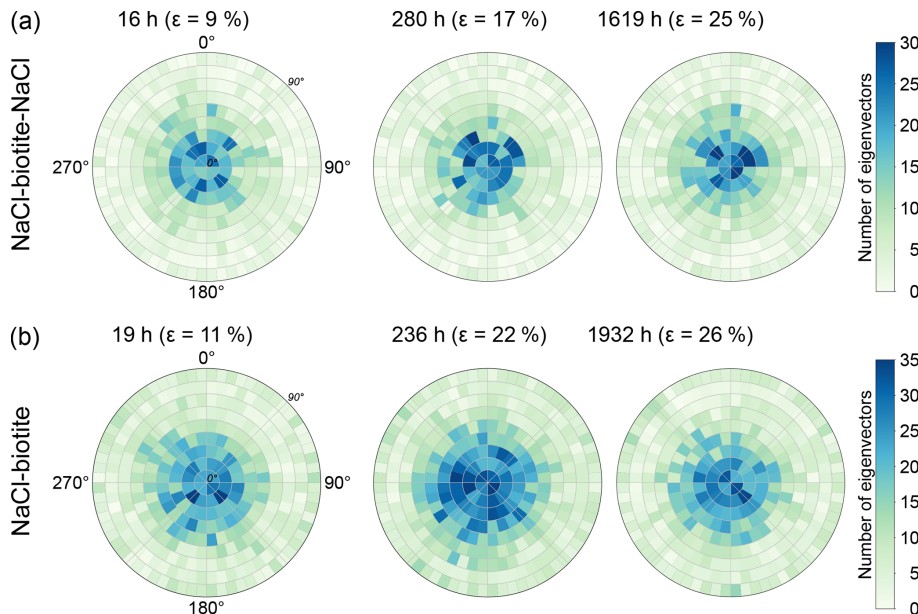

**Figure 7.** Density plots of the eigenvector of biotite grains in **(a)** the SBS and **(b)** the SB sample. The maximum eigenvectors are plotted as densities in Lambert projections perpendicular to the basal planes of the grains. The vertical loading axis is in the centre of the plot. The rings around the central point give the angle between the vertical loading axis and the eigenvector, ranging from 0° in the centre to 90° at the outer margin. High densities of eigenvectors are displayed in blue, whereas lower densities are displayed in green. Note that the biotite grains do not change their orientation significantly with progressing deformation. For both samples, the highest densities can be found in proximity to the central axis indicating that the majority of grains is oriented horizontally within the samples. Numbers of grains measured are from left to right for **(a)** 1836, 1833 and 1986 and for **(b)** 3066, 3836 and 3007, which correspond to 20 wt% of biotite in the respective layer.

served in pure NaCl reference sample. Here, deviatoric and volumetric strain maxima were homogeneously distributed within the bulk sample (Fig. 11). Overall, the dominating character of the volumetric strain was negative in all three samples, indicating compaction.

Furthermore, we calculated average strain rates from the locally resolved strains by dividing each calculated strain value by the time interval length given in seconds. The results are displayed in Table 2 and show an overall decreasing trend in all three samples. In the two biotite-bearing samples, deviatoric strain rate maxima were homogeneously distributed, independent of the layer composition. Comparison of these results to the pure NaCl sample showed no major difference. The observed rates are comparable to the bulk strain rates (see Sect. 3.1.1).

In order to locate the strain maxima in 3-D and compare them to the position of the biotite grains, we plotted the deviatoric and volumetric strain data of the biotite-bearing sample SBS on top of the segmented biotite data (Fig. 12). This showed that in the early stages of our experiment, deviatoric strain maxima corresponded to the location of biotite grains as well as open pore space and pure NaCl clusters. Later on this correlation still existed but was less distinctive. However, local minima and maxima of the volumetric strain did not correlate with the position of biotite grains.

### 3.3 Porosity evolution

A two-dimensional analysis of the porosity in each slice along the loading axis shows that the two biotite-containing samples developed a characteristic pattern in their porosity distribution with progressing deformation. In both samples, the maximum loss of porosity correlates with the location of the biotite-bearing layer (Fig. 13b and c). Whereas in sample SBS the minimum porosity did not fall below $\sim 10\%$ even after 1619 h, we observed the compartmentalisation of SB into a low- ($\phi \leq 2\%$) and a high-porosity ($\phi \sim 15\%$) zone after 236 h (Fig. 13b).

In the pure NaCl sample, the porosity decreased homogeneously and did not fall below $\sim 10\%$ by the end of the experiment (Fig. 13a). Errors for the analyses were determined as 1.7 %–6.9 % for the SBS sample, 1.2 %–8.4 % for the SB sample and 3.8 %–7.3 % for the pure NaCl sample.

### 3.4 NaCl redistribution

Similar to the 2-D porosity, the NaCl content was determined in each slice along the loading axis. The evolution of the relative NaCl distribution in the three samples is shown in Fig. 14. In the SBS sample, the pure NaCl layers gained $\sim 14\%$ and $\sim 16\%$ for the bottom and top layer, respectively (Fig. 14a). A similar trend could be observed in the SB sample (Fig. 14b). Here, the pure NaCl layer showed an

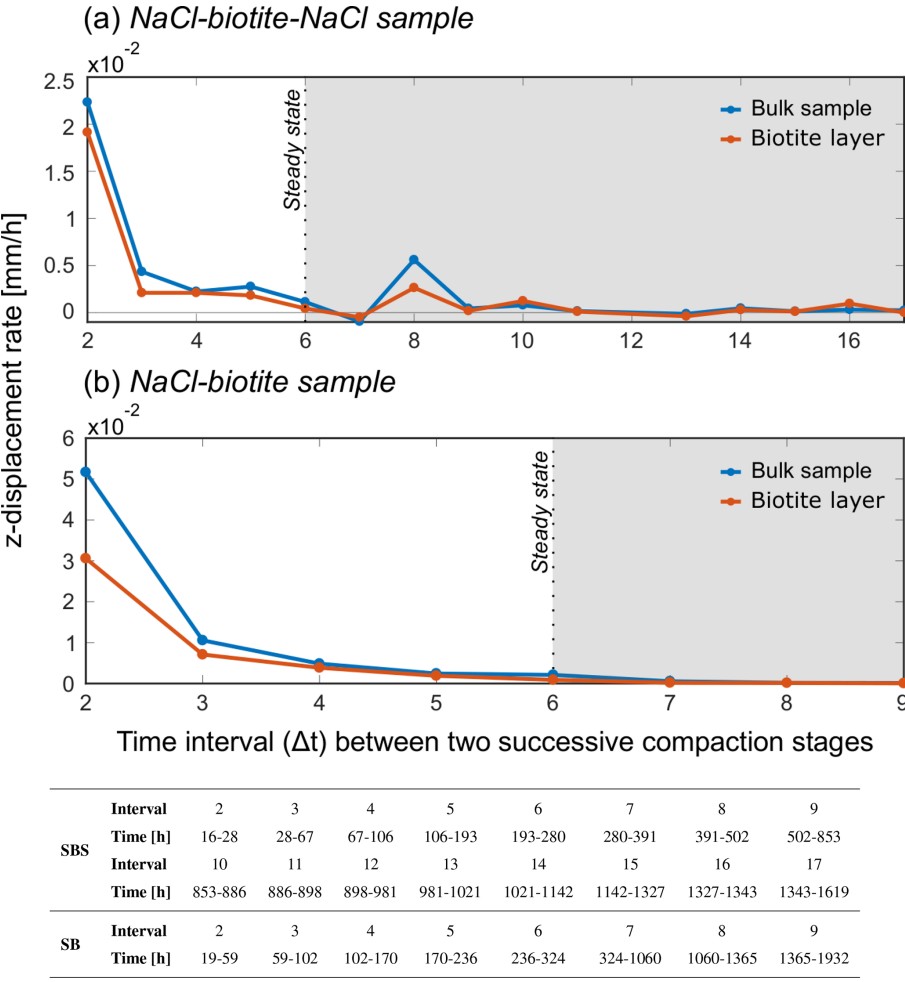

**Figure 8.** The graph shows the vertical displacement rate of the bulk sample and the biotite-bearing layer for different increments of progressing deformation over the duration of the experiments. Panel **(a)** contains data from the SBS sample and **(b)** from the SB sample. Before the apparent steady-state deformation the bulk samples compact faster than the biotite-bearing layer. This trend continues into the apparent steady state (grey-shaded area) although both rates become comparable to each other, indicating that the biotite-bearing layer is not compacting faster than the bulk sample.

**Table 2.** Mean strain rates derived from image correlation.

| Sample | Time [h] | Deviatoric strain rate [s$^{-1}$] | Volumetric strain rate [s$^{-1}$] |
|---|---|---|---|
| SBS | 28–67 | $2.0 \times 10^{-7}$ | $-7.0 \times 10^{-8}$ |
| | 193–280 | $4.8 \times 10^{-8}$ | $-3.3 \times 10^{-8}$ |
| | 1343–1619 | $1.6 \times 10^{-8}$ | $-9.1 \times 10^{-10}$ |
| SB | 19–59 | $3.8 \times 10^{-7}$ | $-3.6 \times 10^{-7}$ |
| | 236–324 | $3.3 \times 10^{-8}$ | $-2.3 \times 10^{-8}$ |
| | 1365–1932 | $7.7 \times 10^{-9}$ | $-3.3 \times 10^{-8}$ |
| S | 18–62 | $2.2 \times 10^{-7}$ | $-1.8 \times 10^{-7}$ |
| | 62–512 | $4.3 \times 10^{-8}$ | $-1.2 \times 10^{-8}$ |
| | 512–1089 | $1.9 \times 10^{-8}$ | $-1.6 \times 10^{-8}$ |

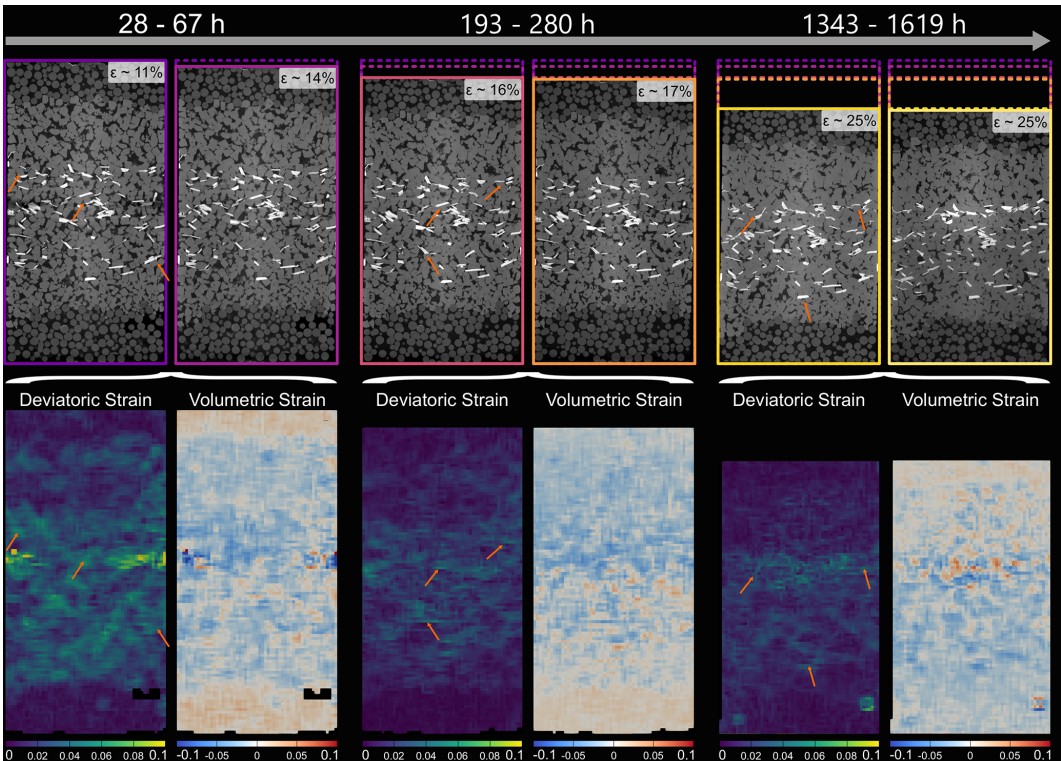

**Figure 9.** Locally resolved strain in the SBS sample. The absolute values of deviatoric and volumetric strain were calculated from digital volume correlation for three different time steps. The top row shows the two microtomography scans which were compared to each other using digital volume correlation. Further, the contours are showing the sample dimensions, ranging in colour from purple for the least compacted sample to light yellow for the most compacted sample. In the bottom row, the results of the strain calculation are displayed. Note that the strains are cumulative and cover the entire time interval between the two selected scans. The durations of the time intervals increase from the left to the right and are 39, 87 and 276 h. For the deviatoric strain, bright colours represent high strains, whereas dark colours represent low strains. For the volumetric strain, blue indicates compaction, while red indicates dilatation. Note that the strains are resolved on the grain scale. Arrows point to high deviatoric strains at NaCl–biotite interphase boundaries.

increase of NaCl by $\sim 13\%$ and the amount of NaCl at the NaCl–biotite interface increased by $\sim 15\%$ (Fig. 14b). Notable shoulders formed at the interfaces between the pure NaCl and the NaCl–biotite layers. In addition to that, no NaCl precipitated in the glass bead layers. The pure NaCl sample on the other hand showed a homogeneous increase of the NaCl content per slice by $\sim 13\%$ throughout the bulk sample. This is comparable to the pure NaCl layer in the SB sample. Errors for the analyses were determined as 1.4 %– 4.5 % for the SBS sample, 2.7 %–5.3 % for the SB sample and 5.8 %–7.5 % for the pure NaCl sample.

Figure 15 shows volumetric analyses that determine the relative proportions of NaCl, pores and biotite in the biotite-bearing layers. In these analyses, biotite acts as an insoluble internal standard. Errors for the measurements were determined from the accuracy of the AI segmentation and are represented by a shaded area around the line plot. Both samples show that, while the biotite content is indeed constant within the analysed subvolumes, the relative reduction of porosity is compensated by a relative increase in the NaCl content in

these volumes. In the SBS sample, just over 10 % of additional salt was measured in the biotite-bearing layer, with a slightly lower increase in the SB sample.

Mass balance analysis (Table 3) shows that no additional NaCl entered the samples; hence, pure redistribution of NaCl was observed in our experiments. The deviation of the mass calculated from binary data is for all three samples in the same order as the absolute error from image segmentation.

## 4 Discussion

### 4.1 Length scales of NaCl transport during DPC – a qualitative exposition

The experiments allowed us to compare bulk compaction of the sample to deformation on the grain scale in three different samples over a total duration of 1089 (pure NaCl), 1619 (SBS) and 1932 h (SB). The general compaction behaviour we observed shows a qualitative similarity to previous studies on NaCl compaction (e.g. Spiers and Schutjens, 1990;

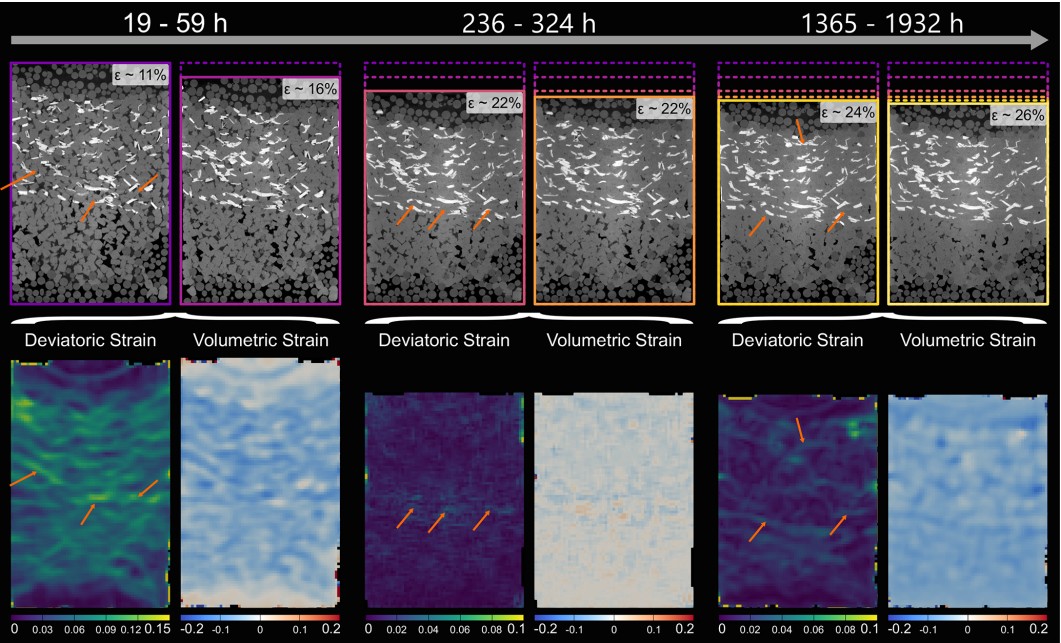

**Figure 10.** Locally resolved strain in the SB sample. The absolute values of deviatoric and volumetric strain were calculated from digital volume correlation for three different time steps. The top row shows the two microtomography scans which were compared to each other using digital volume correlation. Further, the contours are showing the sample dimensions, ranging in colour from purple for the least compacted sample to light yellow for the most compacted sample. In the bottom row, the results of the strain calculation are displayed. Note that the strains are cumulative and cover the entire time interval between the two selected scans. The time intervals increase from the left to the right and are 40, 88 and 567 h. For the deviatoric strain, bright colours represent high strains, whereas dark colours represent low strains. For the volumetric strain, blue indicates compaction, while red indicates dilatation. Note the larger scale for the first time step of the deviatoric strain and that the strains are resolved on the grain scale. Arrows point to high deviatoric strains at NaCl–biotite interphase boundaries.

**Table 3.** Salt content of the samples derived from image analysis and compared to the initial weight by mass balance.

| Time [h] | Mass calculated from binary data [g] | Deviation from initial mass [%] | Abs. error from segmentation [%] |
|---|---|---|---|
| SBS sample | | | |
| 16 | 0.231 | −3.3 | 1.69 |
| 193 | 0.225 | −5.7 | 1.77 |
| 1619 | 0.219 | −8.2 | 4.47 |
| SB sample | | | |
| 19 | 0.176 | 0.5 | 2.91 |
| 170 | 0.164 | −6.4 | 3.76 |
| 1932 | 0.163 | −6.8 | 5.25 |
| S sample | | | |
| 18 | 0.304 | −7.9 | 6.9 |
| 512 | 0.309 | −6.4 | 5.9 |
| 1089 | 0.322 | −2.5 | 5.8 |

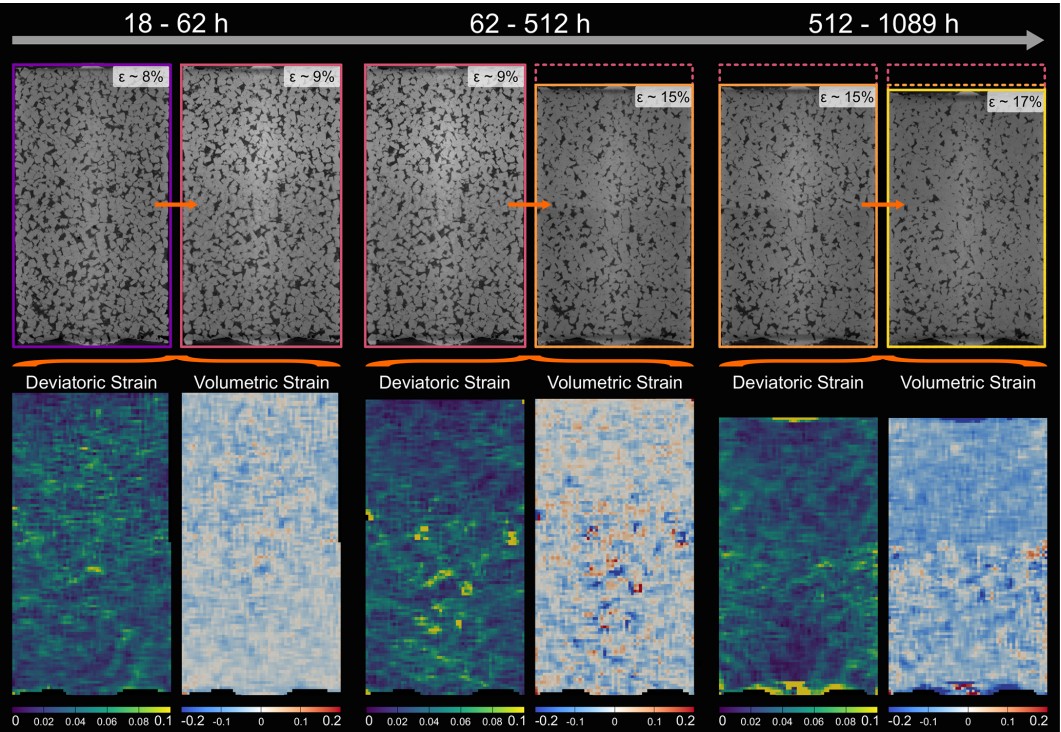

**Figure 11.** Locally resolved strain in the pure NaCl sample. The absolute values of deviatoric and volumetric strain were calculated from digital volume correlation for three different time steps. The top row shows the two microtomography scans which were compared to each other using digital volume correlation. Further, the contours are showing the sample dimensions, ranging in colour from purple for the least compacted sample to yellow for the most compacted sample. In the bottom row, the results of the strain calculation are displayed. Note that the strains are added up and cover the entire time interval between the two selected scans. The time intervals increase from the left to the right and are 44, 450 and 577 h. For the deviatoric strain, bright colours represent high strains, whereas dark colours represent low strains. For the volumetric strain, blue indicates compaction, while red indicates dilatation.

**Table 4.** Starting porosity of individual layers after 1 h of compaction.

|         | SBS                              | SB     | S      |
|---------|----------------------------------|--------|--------|
| NaCl    | top: 30.8 %<br>bottom: 33.0 %    | 25.3 % | 27.0 % |
| Biotite | 30.8 %                           | 24.3 % | –      |

Renard et al., 2004; Macente et al., 2018) except for the attainment of an apparent steady state. The exceptional length of our experiments allows further to qualitatively compare them to compaction data of salt used as backfill material, e.g. at the Waste Isolation Pilot Plant (WIPP) site. However, a direct comparison to published experimental compaction data is difficult as grain sizes and deformation conditions vary between individual studies. Therefore, we used the rate law by Spiers et al. (2004) to calculate strain rates of diffusion-controlled DPC with the parameters used in our experiments (Fig. 16).

At early stages our bulk strain rates are up to 1 order of magnitude higher than the calculated strain rates. However, with decreasing porosity, the measured strain rates progressively approach the calculated rates and broadly align with those. In the early stages of our experiments, elevated compaction rates could be the result of the formation of new effective dissolution sites by grain crushing and microfracturing. We acknowledge that locally plastic deformation may have occurred at small contacts where stress was concentrated; however, the effect upon the bulk deformation was negligible. One reason why DVC works in our experiments is that the centres of the NaCl grains did not deform; hence, deformation must have happened at the grain boundaries and not within the grains. DVC uses, within the limits of noise, the texture of the grains (expressed by the grey-scale distribution) to identify a labelled grain from the reference image in the deformed image. A grain deformed by crystal plastic deformation would show geometrical changes in its grey-scale distribution hence its texture. We would not expect DVC to converge when the loss of mass and the change of grain texture occur simultaneously. We rather explain the differences between measured and calculated strain rates by a mechan-

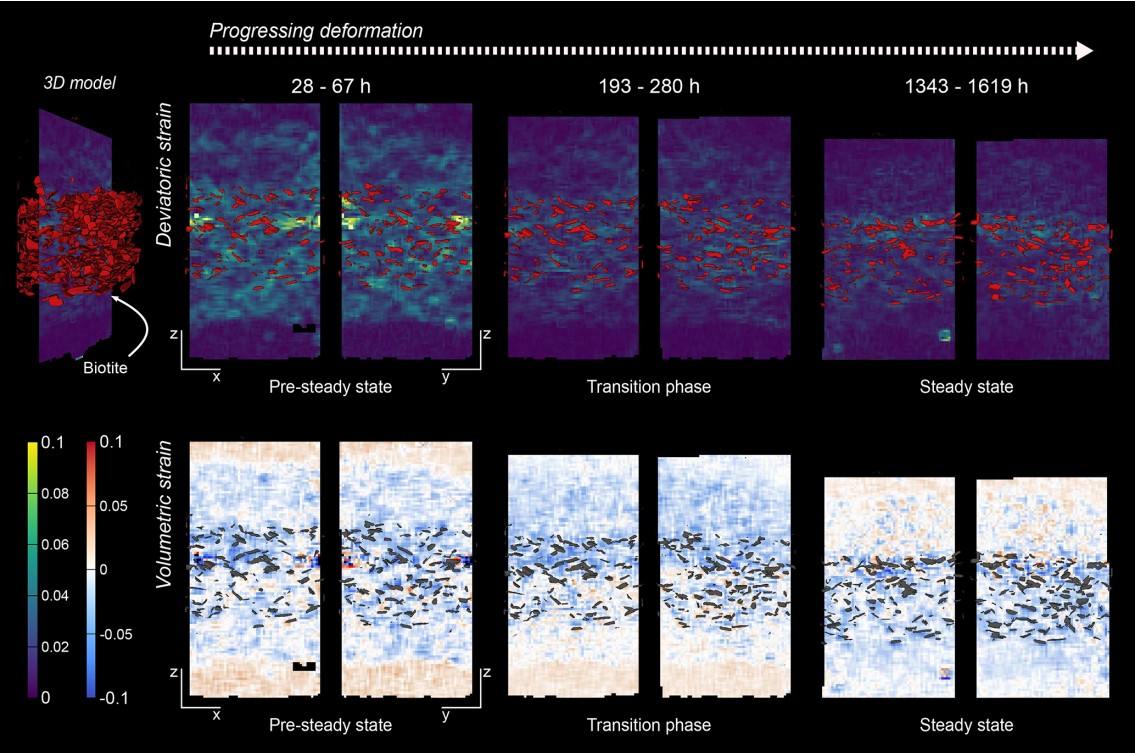

**Figure 12.** Comparison of the location of biotite grains with strain maxima derived from digital volume correlation in the SBS sample. The biotite was segmented from $\mu$CT scans and plotted on top of the corresponding vertical slice through the DVC result. The top row shows the deviatoric strain and the bottom row the volumetric strain of the SBS sample at three different stages of deformation. Note the different orientation of slices to display the three-dimensional information. With progressing deformation, the deviatoric strain is more focused in the biotite-bearing layer. However, maxima also occur in adjacent pure NaCl layers.

ical compaction component in the measured rates which is not included in the calculated rates.

Independent of the sample composition, the compaction was accomplished by an asymptotic decrease of the bulk strain rate. These observations indicate a change in the dominant deformation mode after $\sim$ 250 h of compaction. We interpret the non-linear character of our compaction and strain rate curves to display the transition from a state when mechanical compaction significantly contributes to the strain rate to an interlocked aggregate dominated by chemical compaction. Microstructural observations, which showed enhanced indentation of NaCl grains and reduction of pore space with progressing deformation (Fig. 6), point to the activity of dissolution–precipitation creep (e.g. Rutter, 1983; Tada and Siever, 1989; Gratier et al., 2013) during that period.

Visualisation of our results show that the compaction was accommodated across the entire biotite-bearing samples (Figs. 8–10), which is in conflict with models that postulate a pronounced effect of phyllosilicates on deformation localisation (Heald, 1956; Thomson, 1959; Aharonov and Katsman, 2009) and acceleration of compaction (Hickman and Evans, 1995; Rutter and Wanten, 2000). We tested this observation

by comparing the vertical displacement rates within the sample to the bulk compaction rate and by calculating locally resolved strains using digital image correlation. While we found that the biotite-bearing layers did not compact faster than the bulk samples, we also found that high deviatoric and volumetric strains were not restricted to the biotite-bearing layers. In fact, pure NaCl domains in biotite-bearing samples showed similar strain patterns (Figs. 9–11), and the overall strain distribution in these samples was comparable to the one observed in the pure NaCl reference sample. Therefore, we conclude that deformation was not localised in the biotite-bearing layer but distributed and accommodated across all layers. Macente et al. (2018) came to a similar conclusion, explaining their results with a stress-bearing network of dynamic force chains in the pure NaCl domains which evolve in the granular material as response to a feedback of applied vertical loading and the increase in load-bearing cross-sectional area due to local variations in the dissolution rate (Bruthans et al., 2014). Furthermore, they concluded that deformation was promoted by phase boundaries in the biotite-bearing layer. Similar to Macente et al. (2018), we observed the highest porosity reduction in layers containing phyllosilicates. Compared to the pure NaCl layers, these layers lost

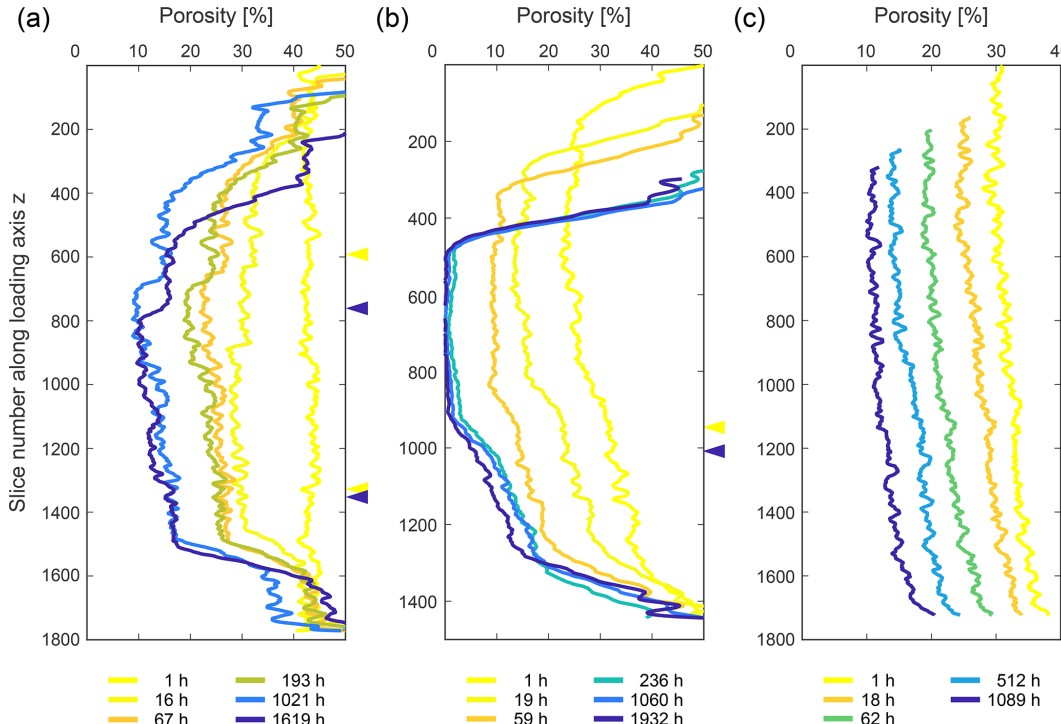

**Figure 13.** Porosity evolution of **(a)** the SBS **(b)** the SB sample and **(c)** the pure NaCl sample with progressing deformation. The porosity was measured as two-dimensional porosity for each slice along the loading axis. Different colours indicate different time steps, ranging from yellow to blue with progressing compaction. While heterogeneities arise in panels **(a)** and **(b)** which show the compositionally layered samples, panel **(c)** shows a homogeneous decrease of the porosity within the sample. The highest porosity loss occurs in biotite-bearing layers, resulting in the compartmentalisation of the SB sample (see Sect. 2.5). Note that the arrows on the right-hand side mark the transition from the pure NaCl domain to the NaCl–biotite domain, and colours correspond to the deformation stage as denoted in the key.

~ 24% more in the SBS, and ~ 41 % more in the SB sample relative to the initial porosity in each layer. Combining the observations of higher porosity loss in the biotite-bearing layers with the evidence that compaction is not concentrated in those layers leads to a paradox that cannot be explained by the classical theory of DPC, which would suggest enhanced DPC to lead to strain localisation and thus a porosity loss. A possible solution would be diffusive material transport from a source outside the biotite-bearing layer into the pore space of that layer.

We were able to show that the NaCl content in the biotite-bearing layers of the SB and SBS sample increased with progressing deformation (see Figs. 14 and 15), and we interpret this salt to have been sourced from the pure NaCl layers. In the SB sample the largest effect can be observed at the layer interface from which a negative gradient emerged towards the top of the biotite-bearing layer (Fig. 14b). In this case, the effect of the NaCl migration is restrained by the breakdown of porosity after 170 h, which limits further migration of dissolved NaCl into the biotite-bearing layer from the suspected source in the pure NaCl layer at the bottom of the sample (Fig. 14b). In the SBS sample, we see no porosity breakdown and no gradient in the NaCl distribution emerge within the biotite-bearing layer. The porosity remains interconnected throughout the entire experiment, maintaining access to both NaCl layers as potential sources for salt. The biotite-bearing layer showed a consistently higher increase in NaCl than the marginal pure NaCl layers (Fig. 14a). The upper NaCl layer especially developed a pronounced gradient towards the interface with the biotite-bearing layer though, which could be evidence for a diffusive salt redistribution. Combined, we consider these observations strong evidence for NaCl diffusion over several hundred micrometres CE3 and multiple grain diameters into the biotite-bearing layer. For an unambiguous quantification of the length scales, further experiments are required which directly trace the dissolved material from source to sink using a tracer technique. However, these must be considered technically extremely challenging and beyond the scope of this study.

Potential driving forces for diffusive transport over extended length scales are gradients in temperature and enhanced solubility by irradiation damage within the samples and between layers of different composition. While we consider the former to be negligible in our samples, we are aware that irradiation damage may occur in our samples and influence the solubility of NaCl. The NaCl which was recovered from the oedometer cells after deformation showed a change of colour from white to yellow, which points towards irradia-

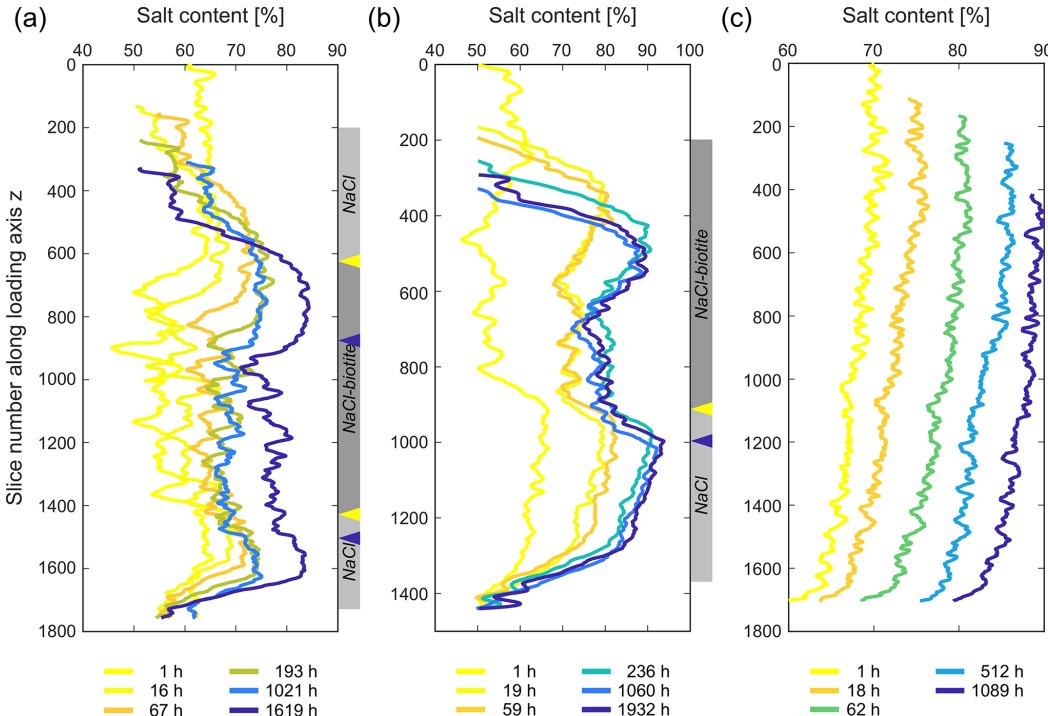

**Figure 14.** Evolving NaCl distribution in **(a)** the SBS, **(b)** the SB and **(c)** the pure NaCl sample with progressing deformation. The NaCl content was measured in 2-D for each slice along the loading axis. Panels **(a)** and **(b)** show that the NaCl content in the biotite-bearing layer increases more than in the rest of the sample, while panel **(c)** shows a homogeneous increase of the NaCl content within the sample. Note that the arrows on the side mark the position of the interface between the biotite-bearing layer and the pure NaCl layer, with their colours corresponding to the time steps as indicated in the key below. The grey bar reflects the sample composition at the initial compaction stage.

tion damage by the formation of F and H centres in the anion lattice of the crystal (e.g. Lidiard, 1998). However, as our bulk samples were all uniformly exposed to a similar dose, we consider lateral damage gradients within the samples to be negligible. Overall, we expect that the doses and irradiation periods in our experiments were too low to report an significant effect of irradiation damage upon the solution rate as even experiments conducted at Synchrotron sources did not observe an effect (e.g. Renard et al., 2004).

A model that could explain our observations of long-distance diffusive NaCl redistribution in the samples was proposed by Merino et al. (1983). They suggest that a locally increased porosity leads to reduced contact areas of grains and consequently higher local stresses along these contacts. In DPC, this causes an increased chemical potential and shift from an equilibrium between the fluid and the salt towards conditions that favour dissolution. Hence, the concentration of dissolved matter in the local pore fluid is increasing and diffusion occurs along a concentration gradient towards domains with a lower porosity, in our samples the biotite-bearing layers. There, due to lower local stresses at the grain contacts, the equilibrium concentration of the solute is reduced and precipitation in the open pore space occurs. This process is self-enforcing and reflects an instability of the system. The driving force of the mechanism is con-

tinuously maintained by the preferential precipitation in the biotite-bearing layer which progressively reduces the local stresses at grain contacts hence, the supersaturation of the pore fluid. For Merino's model to be applicable, the NaCl supersaturations produced by pressure solution must be maintained over transport distances greater than the grain scale before redeposition. Experiments by Desarnaud et al. (2014) demonstrate that, counterintuitively, large supersaturations of NaCl (up to 1.6×) can persist in aqueous solutions before nucleation which explains, inter alia, the absent nucleation and precipitation of NaCl in the glass bead layers. Furthermore, Zimmermann et al. (2015) use atomistic simulations to demonstrate that the nucleation kinetics of NaCl are controlled by desolvation rather than diffusive transport.

We explain the initial porosity variation that is needed for Merino et al.'s feedback model with differences in shapes and sizes between the NaCl and biotite grains, where biotite-bearing layers experienced greater packing density than the pure NaCl layers (Fig. 13), leading to lower initial porosity (Table 4). If the model applies to our results, this would in turn imply that driving forces for transport during DPC even in the absence of advective transport are more diverse than diffusion on the grain scale alone and thus more complex than classical models suggest (e.g. Raj, 1982; Rutter, 1983; Gratier, 1987).

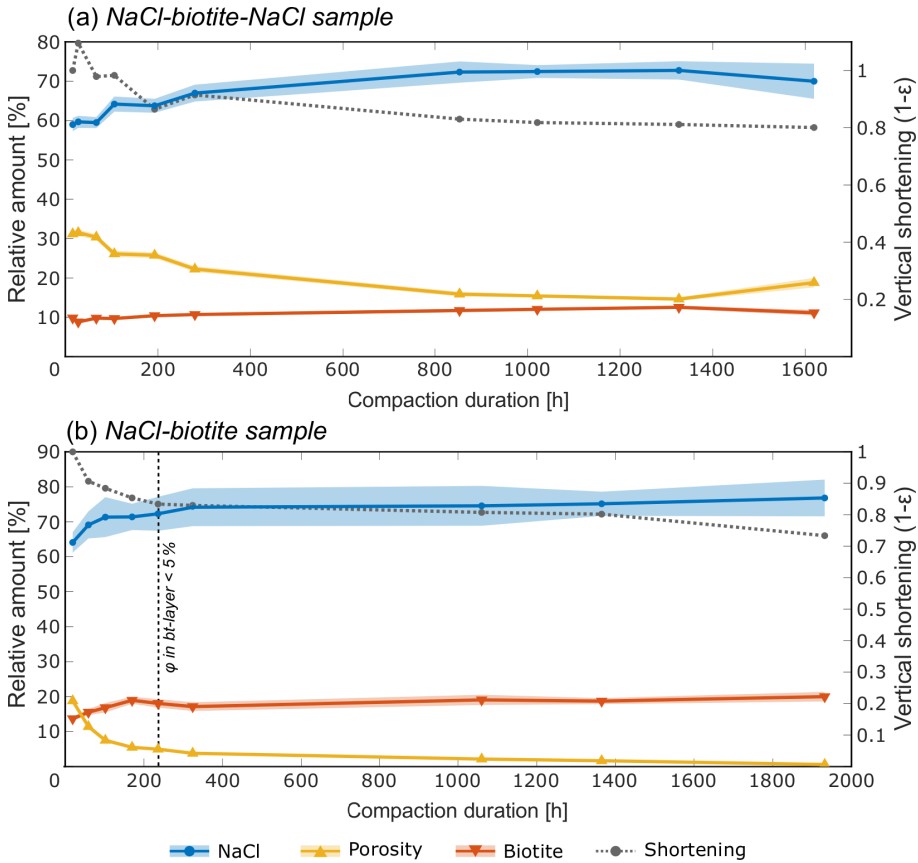

**Figure 15.** Evolution of the relative volumes of NaCl, biotite and porosity in the biotite-bearing layers of **(a)** the SBS and **(b)** the SB sample, with progressing deformation. In both samples, the biotite content remains constant within the segmentation error (plotted as shade in the respective colour), while the NaCl content increases and the porosity decreases. This is persistent throughout the experiment. After 324 h of compaction, the increase of the NaCl content in the SB sample **(b)** stagnates, which corresponds to the breakdown of porosity in the biotite-bearing layer.

This is in accordance with field observations by Heald (1956), Mimran (1977) and Buxton and Sibley (1981), who report observations that challenge the classical theory of precipitation in the vicinity of dissolution sites and invoke larger transport distances in sandstones, chalk and limestones, respectively.

## 4.2 What is the role of the biotite?

Our DVC analysis revealed that a proportion of the maxima in the grain-scale shear strains (see Figs. 9 and 10) corresponded to biotite–NaCl phase boundaries. Such phase boundaries are characterised by significant electrochemical effects (Walderhaug et al., 2006; Greene et al., 2009; Kristiansen et al., 2011), which may accelerate dissolution of NaCl. Visualising such interfaces from our data showed the efficiency of this process (see Fig. 6). At the same time, our label analysis showed that biotite grains did not rotate significantly (Fig. 7). Especially during the early stages of compaction, where the sample still had a high porosity, we would expect point loading to force biotite grains to realign. We in-

terpret the fact that this did not happen as corroborating evidence for the efficiency of dissolution at biotite–NaCl phase boundaries.

An increased efficiency of dissolution along phase boundaries would imply that the biotite-bearing layers should compact preferentially. This is an effect that we clearly did not observe in our data (see previous subsection and Figs. 8–10), which raises the question as to how the preferred dissolution is being balanced. We interpret this to happen in the following way: NaCl that is being dissolved at a biotite–NaCl phase boundary, or also along a NaCl grain boundary in the biotite-bearing layer, is only redistributed locally, within that layer, so that the net volume of that layer is preserved. In the absence of advective transport in our experiment, this is in line with classical DPC theory (e.g. Paterson, 1973; Raj, 1982; Rutter, 1983; Gratier, 1987; Groshong Jr., 1988). This effect is supported by the diffusive potential described above and the additional NaCl that migrates into the layer (see previous subsection), whereby the additional NaCl contributes to

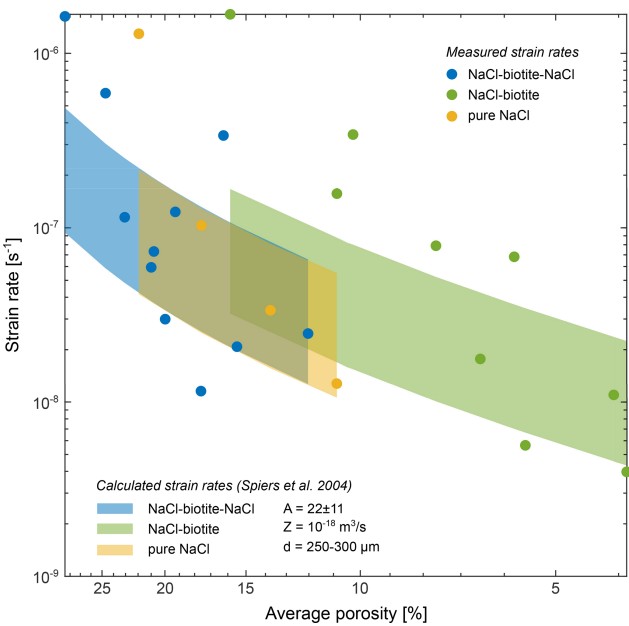

**Figure 16.** Comparison of bulk strain rates and calculated bulk strain rates based on the rate law by Spiers et al. (2004) for diffusion-controlled DPC: $\dot{\varepsilon} = A \frac{\text{DCS}}{d^3} \frac{\sigma_e \Omega^S}{RT} f(\phi)$ with $\dot{\varepsilon}$ as the strain rate, DCS $= Z$ a phenomenological coefficient which represents the effective grain boundary diffusivity, $d$ the grain size, $\sigma_e$ the effective axial stress, $\Omega^S$ the molar volume of the solid, $R$ the gas constant, $T$ the absolute temperature and $f(\phi) = 2\phi/(1 - 2\phi)^2$ as a function of the porosity ($\phi$). For the calculation, the parameters $A$ and $Z$ were used as given in Spiers et al. (1990) and Spiers et al. (2004), respectively. The maximum strain rate was calculated for a geometrical constant $A = 33$ (6-fold coordination) and a grain size of $d = 250\,\mu\text{m}$. The minimum strain rate was similarly calculated for $A = 11$ (12-fold coordination) and $d = 300\,\mu\text{m}$. Our measured data broadly align with the calculated strain rates at the same porosity.

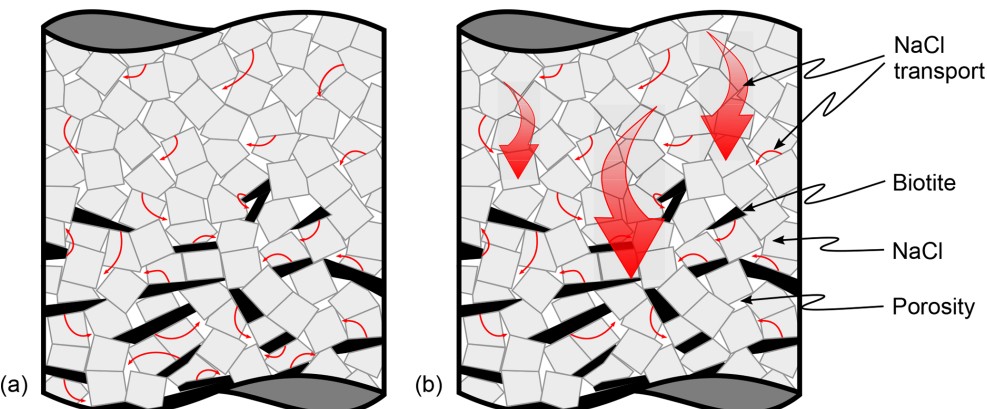

**Figure 17.** Possible transport length scales during dissolution–precipitation creep as proposed within the scope of this work. In panel **(a)**, diffusion occurs only on the grain scale as described in the classical literature. In panel **(b)**, we added diffusive transport on longer length scales as described in Merino et al. (1983). However, diffusive transport on the grain scale is considered to be active as well.

a load-bearing framework whose compaction rate approximates the bulk sample's.

We do note that the biotite composition does not seem to have a first-order effect on pressure solution at its interfaces: while the biotite that Macente et al. (2017; 2018) used had a composition of $K_{0.9}(Mg_{2.5}Fe_{0.4}Al_{0.1}Ti_{0.1})(Si_{2.9}Al_{1.1})O_{10}(OH)_2$ TS3, the one used in this study was richer in iron $K_{0.7}Ca_{0.1}(Fe_{2.1}Mg_{0.3}Ti_{0.2})(Si_{2.5}Al_{1.1})O_{10}(OH)_2$ (see Table B2 in the Appendix for full compositions). Both cation sites are enclosed within the crystal structure of the biotite and are not exposed at the surface of the basal plane. As the latter is likely to be the reactive surface in the dissolution–precipitation creep mechanism, the observed differences in the chemical composition are not expected to affect the dissolution process.

In summary, while biotite grains locally are effective facilitators for DPC irrespective of their composition, it also appears that the chemo-mechanical effect on the entire system is limited and probably outperformed by the trans-domain diffusion outlined above.

### 4.3   A detailed discussion of our DVC analyses

#### 4.3.1   Magnitudes of local strains

Our DVC analyses resolve deviatoric and volumetric strains on the grain scale (Figs. 9–12) and provide insights into the micromechanics of compaction in the various samples. Comparing the strains in the SBS with the SB sample showed the effect of the larger load that was used in the later compaction experiment; the maximum volumetric and deviatoric strains reached in the SB samples are about twice as high.

#### 4.3.2   Character of local strains and their location relative to phyllosilicates

In all three samples, the dominating local volumetric strain was negative. This trend is persistent throughout the experiments and in line with the bulk deformation and vertical shortening of the samples. Deviations from this trend occurred at sites where porosity was reduced by precipitation of dissolved material. Combined, these strains reflect the deformation of the samples by active DPC, tracking the volume changes of the NaCl grains and porosity. The local positive maxima of the volumetric strain (in red) did not correlate with the location of biotite grains but with sites of precipitation (see Fig. 12). However, a pronounced concentration of compaction as described in Macente et al. (2018) cannot be observed.

Deviatoric strain maxima on the other hand showed a certain correlation to the position of biotite grains but not exclusively. Elevated deviatoric strains were also found in the pure NaCl domains in proximity to the biotite-bearing layer which also show elevated strains. It could be argued that deviatoric strain maxima represent the mechanical sliding of grains past each other in an unconsolidated polycrystalline aggregate. However, the process causing the strain maxima continued during the apparent steady-state deformation. We rather relate these strain maxima to grain boundary sliding assisting and promoting dissolution and precipitation processes along grain boundaries independent of the sample composition.

#### 4.3.3   The use of DVC in changing samples

In our analysis, we used continuum DVC to analyse samples that are changing, which violates a core assumption of image correlation, that of the preservation of mass. The fact that material gets dissolved and reprecipitated elsewhere should, in principle, cause problems with convergence of the algorithm. However, across all analysed data sets and increments, the analysis did correlate very well (animation S2 in the Supplement), with only negligible proportions of the volumes not correlating. This observation was persistent across all analysed data.

DVC is very sensitive to the choice of input parameters "half-window size" and "node spacing" which must be adapted to the underlying microstructure – here both of these parameters were carefully tuned to the grain size. Biotite is nonreactive on the scale of our observations and can thus be expected to correlate well. Interiors of NaCl grains are not changing during DPC, as dissolution and precipitation processes occur along the grain boundaries, while the grain centres are not affected by deformation. This means that the algorithm can confidently correlate volumes there. Further, during compaction, the relative proportion of grains versus pores was changing in favour of the former, which will inevitably render the correlation more robust. Lastly, the relative local changes in the texture in the investigated intervals were moderate. Although mass was lost along the grain boundaries, the main body of the particles remains constant over the increment. We consider the combination of these parameters responsible for the successful correlation in our DVC analyses and thus conclude that the application of the technique in our samples is possible.

## 5   Conclusions

In this study, we have investigated active dissolution–precipitation creep in closed NaCl–biotite systems and its contribution to the dynamic evolution of hydraulic rock properties during diagenesis. Our findings indicate that the presence of phyllosilicates enhances porosity reduction but has no obvious effect on deformation localisation, as compaction is accommodated in all layers and local strain maxima are homogeneously distributed within the samples. Following Merino et al. (1983), we explain this paradox with diffusive material migration exceeding the grain scale, from sources in the marginal pure NaCl layer to sinks in the central biotite-bearing layer. This invites a renewed discussion on the influence of phyllosilicates and the driving forces for material transport during dissolution–precipitation creep with direct implications to the diagenetic compartmentalisation of rocks and stylolite formation.

## Appendix A:  Additional information on the methodology

### A1    Experimental conditions for oedometric compaction experiments

Table A1. Experimental conditions for oedometric compaction experiments.

| Sample type | Axial load on actuator [MPa] | Axial load on sample [MPa] | Fluid pressure [MPa] | Effective stress [MPa] | Average temperature [°C] |
|---|---|---|---|---|---|
| Pure NaCl | 0.43 | 6.84 | 0.2 | 6.64 | 22.3 |
| NaCl–biotite | 0.43 | 10.74 | 0.2 | 10.54 | 21.1 |
| NaCl–biotite–NaCl | 0.45 | 7.27 | 0.5 | 6.77 | 21.9 |

### A2    Acquisition parameter for X-ray microtomography scans

Table A2. Acquisition parameter for X-ray microtomography scans. TS4 .

| Scan parameters | SBS | | SB | S |
|---|---|---|---|---|
| | Scans 1 to 8 | Scans 9 to 19 | Scans 1 to 10 | Scans 1 to 5 |
| Acceleration voltage | 100 kV | | 120 kV | 120 kV |
| Target power | 2.8 W | | | |
| No. of projections | 2000 | | | |
| Exposure time | 2 s | | | |
| Source-to-sample distance | 19.5 mm | 18.25 mm | 19.25 mm | |
| Source-to-camera distance | 705.75 mm | 706.25 mm | 712.8 mm | |
| Duration of scan | 67 min | | | |
| Filter | 0.3 mm Al filter | | | |
| 10 dark current and 10 flat field images | yes | | yes | yes |
| Voxel size | 0.005469 mm | 0.005115 mm | 0.005346 mm | 0.005346 mm |

## A3   Location of subvolumes for NaCl measurements

## A4   Lambert projection

The Lambert projection is an equal area projection similar to the Schmidt net used in geology. In contrast to the
Schmidt net, the Lambert projection is a hemispherical polar projection with the pole of the sphere in its centre, while the Schmidt net is a spherical equatorial projection. They are read in a similar fashion. Note that the projections (Lambert and Schmidt) are perpendicular to each other. The interested
reader may be referred to Snyder (1987). We used Lambert projections as they are a good statistical tool to analyse directional data.

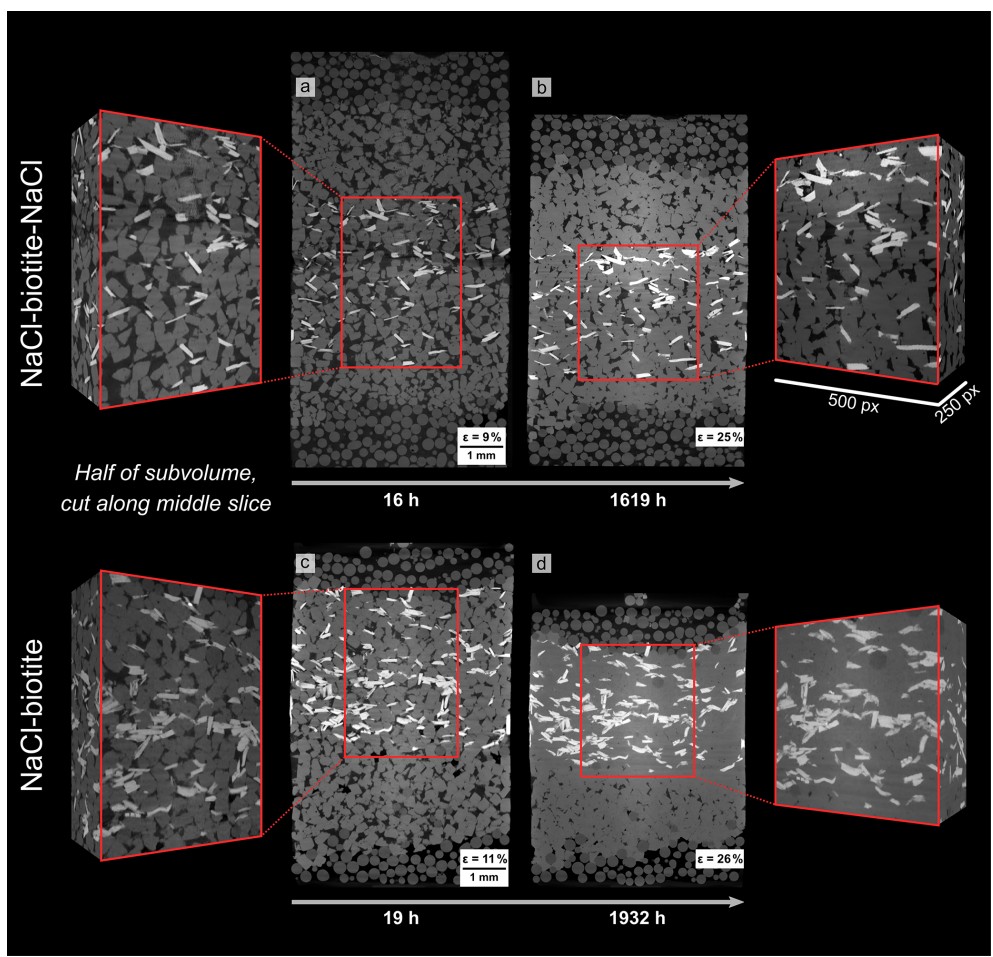

**Figure A1.** TS5 Location of the subvolumes used for the volumetric NaCl measurements relative to the sample. The top row shows the location of the subvolume at the beginning and the end of the experiment within the SBS sample while the bottom row shows it for the SB sample. The subvolumes are centred around a vertical slice that runs through the middle of the sample. Hence, the depicted subvolumes represent only half of the volume used for the measurement.

## Appendix B: Additional information on the results

### B1 Minimum deviatoric strain from DVC measurement

**Table B1.** Minimum deviatoric strain as displayed in Figs. 11, 10 and 9.

| Correlation pair | SB | SBS | S |
|---|---|---|---|
| 1 | 0.00129 | $6.793 \times 10^{-9}$ | 0.00523 |
| 2 | 0.00774 | $8.908 \times 10^{-10}$ | 0.00585 |
| 3 | 0.00494 | $1.499 \times 10^{-9}$ | 0.00441 |

### B2 XRF analysis of biotite

**Table B2.** X-ray fluorescence (XRF) analysis of mica used in our experiments (Bt-BS) and in Macente et al. (2018) (Mc-AM). Values are given in wt% TS6.

| Sample | Bt-BS | Mc-AM |
|---|---|---|
| $SiO_2$ | 35.29 | 42.52 |
| $TiO_2$ | 2.941 | 0.90 |
| $Al_2O_3$ | 12.60 | 14.30 |
| $Fe_2O_3$ | 35.68 | 6.28 |
| $MnO$ | 0.381 | 0.06 |
| $MgO$ | 2.99 | 24.19 |
| $CaO$ | 1.32 | 0 |
| $Na_2O$ | 0.07 | 0 |
| $K_2O$ | 7.647 | 10.39 |
| $P2O_5$ | 0.091 | 0.01 |
| LOI | 0.81 | 0.77 |
| Total | 99.83 | 99.42 |

*Data availability.* The data that support the findings of this study are available from the corresponding author, BS, upon reasonable request.

*Supplement.* The supplement related to this article is available online at: https://doi.org/10.5194/se-12-1-2021-supplement.

*Author contributions.* BS, FF and IBB designed the study. BS ran the experiments. EA supplied the digital volume correlation code and helped with the analysis. All authors were involved in the interpretation of the results and the writing of the final manuscript.

*Competing interests.* Florian Fusseis is topical editor of SE.

*Acknowledgements.* We would like to thank Johannes Glodny from GFZ for the provision of sieved biotite mineral separates. Alex Hart and Ivan Febbrari are thanked for technical support. We also kindly thank the two anonymous reviewers for their helpful reviews.

*Financial support.* This work was financially supported be the School of Geosciences, The University of Edinburgh.

*Review statement.* This paper was edited by David Healy and reviewed by two anonymous referees.

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

## Remarks from the language copy-editor

CE1     Please check and confirm.

CE2     For Figs. 5, 6, and 9 please clarify exactly what changes should be made. Are the words cut off too close to the edge? All the panels have a black background, so which "black panel" should be centered?

CE3     Please check and confirm.

## Remarks from the typesetter

TS1     Please check change.

TS2     Please check change.

TS3     Please check.

TS4     Please note, we have to ask the handling editor for his/her approval to insert the new values here. Please send an explanation why these corrections should be inserted. Thanks.

TS5     Please accept our guidelines to arrange figures in the Appendix here. Thanks.

TS6     Please check change.