# Peer review of "Biotite supports long-range diffusive transport in dissolution-precipitation creep in halite through small porosity fluctuations"

_Solid Earth, 2021_

## Author Comment (AC1)

**Response to Referee no. 1:**

Dear Referee no. 1,

we would like to thank you for your very accurate and constructive revision of our manuscript. We appreciate the time and effort that you and referee no. 2 have dedicated to providing your valuable feedback on our manuscript. We are able to include most of your corrections and suggestions and are sure that they will improve our manuscript. The changes will be highlighted in the revised manuscript.

Please find below our detailed responses to your individual comments.

(*Reviewer comment*; Author's reply)

*1) Title : I find the title misleading as it suggests that the observed effect is caused specifically by biotite. However, the authors seem to argue that it is not specifically the biotite that causes the observed effect but initially (?) reduced porosity in the biotite-bearing layers. If they are right, then a biotite-free but denser salt layer in the sample would show the same layer-scale mass transport phenomenon reported here and described by Merino et al. I would recommend a title more along the lines of "CT imaging demonstrates interlayer mass transport in layered halite-biotite aggregates undergoing dissolution-precipitation creep".*

> We agree with the reviewer's suggestion. The original title suggests indeed that biotite has a specific impact upon the transport length scales of DPC. In the manuscript we argue that the observed effect is rather a consequence of textural heterogeneity and porosity fluctuations between biotite-bearing layers and pure NaCl layers. However, as we discuss in comment 4, the differentiation between the phyllosilicates involved as well as the rate controlling process seems to be important when discussing phenomena related to DPC. Therefore, the revised manuscripts will be titled "Biotite supports long-range diffusive transport in dissolution-precipitation creep in halite through small porosity fluctuations"

*2) Abstract, lines 5-6 read: "We used time-resolved (4D) microtomographic data to capture the dynamic evolution of the transport properties in layered NaCl-NaCl/biotite samples". This is not true. No attempt was made to calculate transport properties (or measure them). Only porosity evolution was studied. Best correct to porosity rather than "transport properties" – throughout the ms.*

> Thank you for pointing this out. The reviewer is correct as transport properties are by definition: conductivity, diffusivity, and viscosity. In our study, we did not measure those. Accordingly, throughout the revised manuscripts, the term "transport properties" will be replaced by "porosity" or "hydraulic properties" where a more general view is emphasized.

*3) Abstract, lines 12-14 reads: "We propose that, in our experiments, the diffusive transport processes invoked in classical theoretical models of DPC are superseded by chemo-mechanical feedbacks that arise on longer length scales." This cannot be said if in the main text it is claimed that the sample scale compaction behaviour is consistent with compaction experiments on pure NaCl. The effect of interlayer transport in the present experiments is argued not to influence overall compaction strain,*

*so it does not dominate over pressure solution as a deformation mechanism, it merely contributes and dominates porosity reduction in the biotite-bearing layers.*

> We agree with the reviewer and will replace "superseded" with "complemented" or an equivalent synonym in the revised manuscript to address the contributing character of the process.

*4) Introduction, lines 29-30 read: "Phyllosilicates have been recognised to have a reinforcing effect on the dissolution process ..".  Yes, a but others have notes that pressure solution (compaction) can be inhibited or unaffected by phyllosilicates, e.g. Niemeijer & Spiers (2002). The enhancement effect comes mainly from observations on natural rocks where advective mass removal along phyllo-rich layers cannot be eliminated as playing a role.*

> We agree that this is a crucial question that needs to be addressed. However, we think that a detailed discussion is beyond the scope of this manuscript and may require another study or publication that focuses on the effect of different types of phyllosilicates upon the three types of DPC systems (dissolution-, diffusion- or precipitation-controlled).  A general statement like "all phyllosilicates have the same effect upon all types of DPC systems" is probably too broad and needs to be tested in a series of scientific experiments.

> As pointed out by the reviewer we agree that observations from natural samples always include an effect of advective flow and mass transport. However, experimental studies by e.g. Renard et al. (2001) show as well that the presence of clay enhances the DPC rate in a closed NaCl system. As DPC in NaCl is diffusion-controlled (see Spiers et al. 1990.) it makes sense that a phyllosilicate that can incorporate an additional layer of water into its structure (like clay minerals can do) provides a good diffusion pathway, hence, accelerates the DPC rate.
> In contrast to the diffusion-controlled NaCl system, Niemeijer & Spiers (2002) investigated muscovite + quartz which is under the experimental conditions of 500°C identified as a dissolution-controlled system. Here, the authors argue themselves that $Al^{3+}$ is expected to decrease the solubility of quartz hence, impede DPC. It would be interesting to see if a low Al mica like biotite has the same effect upon DPC in quartz.

*5) Intro, lines 30-35: What does the present study actually add to the paper by Macente et al (2018)?  Would be wise to make this clear somewhere, e.g. in lines 46-48.  Just seems like a technical refinement at present.*

> While we appreciate the reviewer's feedback, we respectfully disagree that this study is just a technical refinement of Macente et al. (2018). We think this study makes a valuable contribution to the field because it uses technical advances in form of an evolved experimental setup as well as advanced analysis techniques and codes. Analyses that distinguish out study from Macente et al. (2018) only became available recently and allow for example to track the NaCl redistribution. In addition to that, while our study focuses on the qualitative analysis of transport length scales during DPC, Macente et al. (2018) emphasize the impact of phyllosilicates upon the evolving porosity.

> Scientific progress sometimes happens by groundbreaking discoveries but most of the time carefully planned small steps prepare the ground for innovation. We acknowledge that our study may fall into the latter category however, even testing the fundamental observations

by Macente et al. (2018) and confirming the results should be regarded as a successful step towards understanding the process of DPC. We believe that a problem exists in geosciences which is that experiments are often run once and never again. Hence, most discoveries lack replication and confirmation by subsequent studies either through technical difficulty, expense, or because of data handling limitations.

*6) Intro lines 48-49 read: "Our aim was to determine length scales of diffusive transport in a dynamically evolving porosity during DPC". What about trying to explain them??*

We will rephrase the sentence in the revised manuscript and will add an explanatory part.

*7) Section 2, line 53. Peach and Spiers 1996 is a study of the percolation threshold in dilating salt, not a study of deformation mechanisms. A far more relevant reference here and in line 57, would be the study of pressure solution in compaction by Spiers et al 1990, which specifically addresses the creep law for pressure solution in NaCl in 1D compaction and deviatoric creep – and emphasizes the analogue aspect.*

Thank you, for pointing this out. We have changed the reference accordingly as we appreciate the relevance of Spiers et al. 1990 in this context.

*8) Section 2, lines 58-59 read: "It is further a material used in geological nuclear waste repositories (Powers et al.,1978), and its deformation behaviour is well-characterised". Salt is not a material used in radioactive waste repositories – it has been and still is widely considered as a HOST ROCK for repositories. A more recent ref than Powers should be added and refs should be added to underpin "well characterised". Urai , Schleder, Spiers and Kukla 2008 would be suitable here.*

We agree with the reviewer that salt is considered as a host rock for nuclear waste repositories rather than it is a material used in these repositories. The sentence will be rephrased to "It is further considered as a host rock for geological nuclear waste repositories (citations) and its deformation behaviour is well-characterised (citations)". Further, relevant references will be added.

*9) Section 2.3 Experimental setup, lines 95-96 reads: "The experiments were run inside a thermally insulated box where the temperature was logged and found to be stable within ± 1.7 °C over the course of the experiments." This is quite a large range in T for such a soluble material as NaCl (which would certainly cause sample-wide dissolution-precipitation effects) and raises questions regarding temperature GRADIENTS in the sample and their possible effect on convection and advective transport. Was temperature measured at different points along the length of the sample and if so what was the T profile or gradient? Could this have driven advective transport in the samples? Some calculation is needed to answer this. Of further interest here is the possible effect of differential heating of the sample during CT-imaging, as a result of x-ray attenuation – e.g. differential heating of biotite-bearing versus pure NaCl layers. Can effects such as this be eliminated?*

Thank you for pointing this out. Although we agree that this is an important consideration, it is not appropriate for inclusion in this manuscript because the solubility of NaCl is relatively unaffected by the temperature in the range of ± 1.7 °C at room temperature (see figure

below). The solution of NaCl is only a very low endothermic reaction, therefore temperature does not affect the solubility as it would e.g. KCl.
In addition to that, we consider the peek cell surrounding the samples as an additional thermal insulation layer due to its low thermal conductivity ($0,25$ W m$^{-1}$ K$^{-1}$). Hence, temperature changes inside the sample cell are expected to be below 1 °C and therefore negligible for changes in the solubility of NaCl. Unfortunately, we were not able to measure a temperature gradient along the length of the sample but following up from the argumentation above, a measured gradient would have been too low to drive convection or advective transport.

Differential heating of layers with different composition is a function of the dose and the specific heat capacity of the minerals involved. In the case of NaCl and biotite the specific heat capacity is c=0.88 kJ/kg*K and  c =1.035-2.064 * 10^4 kJ/kg*K, respectively. From a theoretical point of view this may cause differential heating by x-ray radiation hence, a temperature gradient within the sample between layers that contain biotite and those that do not.
However, even at high doses at synchrotron facility the increase of temperature of geological samples is in the range of single digits. We would expect to see an even lower effect in lab based x-ray CT scanners. The x-ray source used for the scans was operated with a target power loading of 2.8 W. Compared to e.g. the Diamond Light Source (I12) which at 53 keV and 300 mA ring current would be about 16 kW, our source is a dim heat source. In addition to that, the biotite-bearing layers contain 20 wt% of biotite, are therefore dominated by NaCl which itself will exchange heat with neighbouring biotite-grains. The temperature gradient that can be achieved between the mixed and the pure NaCl will be limited and negligible.

[Figure]

**Solubility Curves**

(From: CK-12 Foundation – Christopher Auyeung)

*10) A further point related to the above is the issue of radiation damage and its effect on NaCl solubility. Recent measurements that I have witnessed in a similar scanner show heating of NaCl by a few degrees accompanied by significant radiation damage of the salt – it turns yellow or purple at high doses. So my question to the present authors is: did the samples change colour after CT scanning?  Did they check?  And, if the colour did change, can they eliminate the possibility of damage gradients influencing dissol-precip transfer between layers of different composition hence different damage in the NaCl?  Note that from a theoretical point of view, if the deposited energy due to radiation damage of NaCl is E, the increase in solubility for small E is 100.E/RT %. Could this effect, or the heating due to attenuation, be significant?*

Thank you for pointing this out. We are aware that radiation damage may occur in the samples. The described change in colour was observed in NaCl recovered from the sample cells. Source of this changes in colour, from white to yellow or even purple, is the formation of F- and H- centres in the anion lattice of the crystal. As the bulk sample experienced a similar dose, we do not expect a lateral damage gradient to emerge and drive mass transfer from higher damaged domains (increased solubility of NaCl) to lower damaged domains. We would also like to point out, that the experiments conducted by Renard et al. (2003) at a synchrotron source do not report any significant effect of radiation damage on the solution rate. Compared to these data, our samples were irradiated at much lower doses and only for a short amount of time (1hr).

We will acknowledge the problem of irradiation damage with an addition to our revised manuscript.

*11) Also under Section 2.3, it is mentioned in line 98 that the applied effective stress on the compaction experiments was 6.64 to 10.5 MPa. That means that local stresses at NaCl and NaCl-biotite grain contacts would have been much higher – in the range 12 to 50 MPa. These stresses are well inside the regime where salt deforms plastically at room T, leading to a coupling between work-hardening plasticity on the grain scale and dissolution-precipitation transfer, as opposed to classical pressure solution seen in compacting NaCl at stresses below 3 -4 MPa (see Urai et al 2008 above; also Spiers & Brzesowsky. Densification behaviour of wet granular salt: Theory versus experiment. Seventh Symposium on salt 1, 83-92, 1993). The likelihood that this plasticity-coupled mechanism played a role in the present experiments, rather than classical pressure solution, should be pointed out, especially as it is a mechanism where pore volume diffusion plays a role as opposed to the grain boundary diffusion process that controls "normal" pressure solution.*

We appreciate the reviewer's feedback and will acknowledge this limitation in our revised manuscript. Although, we agree that locally plastic deformation may have occurred at small contacts where stress was concentrated, the effect upon the bulk deformation was negligible. One reason why DVC works in our experiments is that the centres of the NaCl grains did not deform, hence deformation must have happened at the grain boundaries and not within the grain.

*12) Section 3 Results, Figure 3. The apparently straight portion of the compaction curves shown in this plot is referred to by the authors as steady state creep, whereas the inset in the Fig clearly shows that the strain rate is continuously decreasing within the resolution of the data. Moreover, the authors actually say that the compaction curves show asymptotic behaviour (e.g. line 271), which in itself means that steady state is not achieved. In addition, it is quite impossible to reach a steady state compaction rate in a compaction experiment of any kind, as porosity is continuously decreasing and therefore so must the strain rate – regardless of deformation mechanism. In this study, apparent steady state seen in the compaction curves is an artifact of the few, rather scattered strain-time data (clearly understood from the inset in Fig 3). Perhaps use of the term "apparent steady state" would be acceptable, but the term steady state creep should be removed throughout and all related points corrected accordingly.*

As suggested by the reviewer we have corrected in the revised manuscript the term "steady state" to "apparent steady state".

*13) Figure 7. Lines 223-224: "Figure 7 shows the vertical displacement rate of the biotite-bearing layer and the bulk sample for different increments of progressing deformation". And in Lines 225-226 "At the beginning of the experiment the rate of both bulk samples was elevated compared to the biotite-bearing layers." OK for the displacement rates, but any meaningful comparison requires normalization with respect to the thickness of the NaCl and NaCl-biotite layers considered, i.e. the average strain rates in each zone should be plotted versus compaction stage (time proxy). This is crucial because of later discussion around the issue of enhanced compaction (lower contact stresses) causing interlayer mass transfer.*

We chose to plot the vertical displacement rate instead of the average bulk strain rate as the latter would give a similar result to the DVC analysis. While the average bulk strain rate of a layer would describe the magnitude of strain, DVC gives strain resolved on the grain scale which we believe is a better measurement to analyse strain localisation and enhanced compaction.

*14) Section 3.2 Strain analysis. The usage of the terms volumetric strain (isotropic), deviatoric strain and compaction strain becomes a bit confused from here on, I feel. In 1-D, compaction strain is equal to volumetric strain, but not equal to the isotropic strain component of the strain tensor of course. However, the isotropic vol strain does seem to be referred to as compaction at some points in the ms. Somewhere early in the ms, these terms need to be strictly defined and differentiated from each other, and then used consistently. It is also important to note that deviatoric strain cannot occur during 1D compaction independently of the isotropic component of volume reduction, because the pressure solution process (even when accompanied by plasticity) is serially coupled to intergranular sliding – you cannot have one without the other (in pure NaCl or in NaCl-biotite mixtures). In isotropic compaction under 3D loading with S1=S2=S3 you can get compaction with little or no intergranular sliding.*

Thank you for pointing this out. We agree that these terms need to be defined in the manuscript. Accordingly, we will add those changes to the methods section. We further emphasize the differentiation between macro- and micro-strain. While the vertical shortening and compaction of the bulk sample refers to the macro-strain, isotropic (volumetric) and deviatoric strain refer to strains resolved on the grain scale. Here, the volumetric strain is the first invariant and isotropic strain component of the strain tensor. The deviatoric strain is equal to the second invariant of the strain tensor and describes deformation at constant volume.

We would like to point at that the term „compaction strain" was not used in our manuscript.

*15) The above point comes into play in Figs 8-11, where isotropic volumetric strain (called volumetric strain) is used as an indicator of compaction, whereas macroscopically measured compaction is 1-D compaction. I would strongly advise the authors to present a complete picture in Figs 8-11 by adding contour plots of vertical compaction strain, in addition to the isotropic vol and deviatoric strains. This would make what is going on clearer with a complete set of all information.*

As suggested by the reviewer we will adjust figures 8-11 by adding a vertical shortening component.

*16) Section 3.4 NaCl redistribution, Fig 13 and text referring to it (e.g. lines 254-255). Here, changes in NaCl content within the samples are specified per horizontal slice through the sample. That should be made clearer in the text as it reads as though the mass of the samples is not constant. That also raises the question as to whether the mass of NaCl in the samples is indeed constant. Do the changes in NaCl mass/vol fraction seen in individual samples add up to the original NaCl solid mass? This needs to be clarified.*

We agree with the reviewer and will rephrase the text to clarify the case. An increase in NaCl could indeed be an addition of NaCl from an outside source. This is not the case in our samples. Instead, the total NaCl content of the samples remains constant throughout the experiment.
We will add a measure of the total NaCl in the samples and give mass balance calculations to prove that no additional NaCl enters the sample and the pure redistribution of NaCl was observed.

*17) Section 4 Discussion, lines 269-270 read: "The general compaction behaviour we observed was consistent with previous studies on NaCl compaction". Well, yes, the data do show increasing compaction with time. But that is no basis to claim consistency with previous work. First, no other compaction data on salt show the apparent steady state portion claimed by the authors, so they are not qualitatively consistent. Second, a comparison with the isostatic compaction tests of Schutjens & Spiers is not expected to be consistent because of the different boundary conditions imposed. Third, no evidence is presented that the present amounts and rates of compaction are consistent with previous 1D compaction tests on samples of controlled grain size, such as those reported by Spiers et al (1990 – low applied stresses) or Brzesowsky and Spiers (1993 – stresses similar to the present). To claim any consistency or detect any interesting differences, a quantitative comparison should be made by adding a few curves from previous 1D compaction studies on salt of the same grain size – or calculating compaction curves for the present conditions from the compaction data or laws given by previous authors.*

We agree that claiming "consistency" of our results with previous work requires a proof which we do not establish in the original manuscript. In the revised manuscript we will therefore exchange the term "consistency" with "in accordance with".

A direct comparison of our results with previous studies is difficult as grain sizes and deformation conditions vary between the individual studies. According to the rate law by Spiers et al. (2004) the strain rate in diffusion-controlled systems is affected by the grain size, the applied effective load, the temperature and the porosity. Further, the duration of the here presented experiments exceeds the duration of most published data.
Hence, compaction curves could be calculated based on the rate law by Spiers et al. but previously published data should not be plotted for direct comparison.
We will follow the suggestion by the reviewer and add a calculated compaction curve to our data plots in the revised manuscript.

*18) Lines 272-276. The authors claim a change in deformation mode beyond 200 hours here. But they also argue that their data are continuous and show a continuous asymptotic decrease in strain rate. The continuous nature of their strain rate data is also apparent from Fig 3 (see point 12 above). It does not seem justified then to claim a change in deformation mode here, so the point should be removed or weakened.*

We agree with the reviewer, the two stages of the process are rather related to a transition from a loosely packed aggregate where mechanical compaction significantly contributes to the strain rate, to an interlocked aggregate dominated by chemical compaction. In this case mechanical and chemical compaction are active simultaneously throughout the experiment but dominate at different times depending on the state of the aggregate. We will add a clarification to the revised manuscript.

*19) Lines 305-306 read: "the upper NaCl layer did develop a pronounced gradient towards the interface with the biotite-bearing layer though, which could be evidence for a diffusive salt redistribution". Yes agreed. But it could also be evidence of advective redistribution if there were even small internal T-gradients. Can this possibility be eliminated? If not that should be stated.*

Thank you for pointing this out. As discussed for comment 9 we think that internal temperature gradients are negligible.

*20) Lines 309-319: Here it is proposed, quite reasonably, that Merino's model of diffusion from more porous to denser layers may occur because of a higher solute concentration (supersaturation) in the more porous material than the denser material. This is consistent with pressure solution theory and fine. However, appealing to the high supersaturations discussed by Desarnaud et al (2014) or Zimmerman et al (2015) is misplaced here as these are concerned with pre-nucleation supersaturations. There is no evidence for a nucleation stage in the present experiments as it is quite clear from the grain scale images, and from previous compaction work on NaCl, that precipitation occurs mainly by OVERGROWTH on the pre-existing grain (pore) walls. If fine grains are nucleated in the pores in the present experiments, that would be new and should be described. Only then should the above nucleation argument can be kept.*

We appreciate the reviewer's feedback, but we respectfully disagree with the suggestion to remove the argument.

The reviewer's statement that NaCl precipitation occurs by overgrowth as a simplistic process, ignores the fact that crystal growth mechanisms are varied and largely subject to supersaturation. In fact, the resolution of the CT data does not enable us to make any statement about growth mechanism at the NaCl surface. We chose these references as they confirm that it is possible to achieve a significant supersaturation of NaCl in solution without homogenous nucleation (which we do not observe). This is a prerequisite for long distance transport.

We agree that long distance transport is consistent with DPC theory (e.g. Gunderson et al 2002), but our argument, as presented, indicates that such behaviour is consistent with the specific material studied as well as theory in general.

*21) Lines 320-322. Here the authors argue that the Merino model may apply because the biotite-bearing layers compacted more than the pure NaCl layers in the early stages of the experiments, so had lower porosity, lower contact stresses and hence a lower supersaturation on NaCl in the pores – giving a driving force for diffusion of dissolved NaCl from the pure to the mixed layers. For the reader, however, this seems to be a strange statement after so much emphasis has been placed on the lack of evidence for any strain enhancement in the biotite-bearing layers (at many points, but also again in lines 336-337). The argument seems inconsistent. Can the authors please clarify this picture – it is*

*most confusing in the present form???? Was strain only uniform in the late stages but not initially? If so, please make this clearer.*

> We will clarify this in the revised manuscript. The inconsistency and confusion results from an error in the wording. Instead of a higher compaction, the packing density of the biotite-bearing layers was greater than that of the pure NaCl domains. Hence, the porosity was lower. We suggest adding the initial porosity measurements in the results section to demonstrate that the lower porosity existed right from the beginning and is not an artifact of localized compaction in the biotite-bearing layer.

*22) In relation to the above point, I also wonder if the authors should mention the possibility that the preferential "cementation" of the biotite rich layers that they see could reflect an INSTABILITY caused by the Merino effect progressively reducing grain contact stresses and supersaturation in the biotite layers faster than in the NaCl layers.*

> We will add this to the revised manuscript. We assume the reviewer means an instability that is the reinforcing counter part to the stress increase in high porosity domains in the Merino feedback mechanism.

*23) Line 331. The authors suggest here that electrochemical effects at the NaCl-biotite interface may enhance dissolution at those sites, following the references cited. However, as far as I recall those refs deal with the effects of micas at mica-quartz interfaces. I do not think one can then assume that the same enhancement effects will occur at a mica-salt (ionic solid) interface. Line 331 should read "…..which MAY accelerate dissolution of NaCl.".*

> Line 331 will be updated as suggested. It is correct that the cited literature deals with mica-qtz interfaces. However, the authors already argue themselves that their results may be applicable to dissimilar interfaces in general.

*24) Lines 336-334: This explanation of what goes on inside a biotite-bearing layer is reasonable. However, is it not a remarkable coincidence that "the additional NaCl contributes to a load-bearing framework whose compaction rate is in sync with the bulk sample's"??? Would it not be better (i.e. more accurate) to replace "is in sync with" by "roughly matches" ?? Otherwise there would have to be some strong coupling which is hard to argue.*

> As suggested by the reviewer we have replaced "is in sync with" by "approximates".

*25) Lines 374-386. The issue of volumetric strain versus compaction strain versus isotropic strain raises its confusing head here again, further underpinning the need for better definition of these terms at an early stage in the paper, followed by consistent use in a way that distinguishes between physical compaction and the math properties of the isotropic part of the strain tensor – see point 14 above. MORE INTERESTING though is the issue of what was observed in the glass bead layers in the biotite-bearing samples. Presumably there was no actual compaction of these layers, beyond some rearrangement effects or possible bead breakage or chipping. This should be clarified in the Results. There, it should also be made clear whether there was any precipitation of salt in the bead layers. If there was at both sample ends, this would support the Merino model, as there would be no*

*stress-induced supersaturation in the brine in the pores between beads. If there was precipitation between beads at one end of the sample but not the other, this would suggest a role of convection and advective transport, or double diffusive convection. If there was no precipitation at all between the beads, this could be explained by the nucleation barrier at these sites – thus supporting neither the Merino model nor an advective transport model.*

As suggested by the reviewer we will add a paragraph about the glass beads in the result section. In the SBS sample which kept a connected porosity throughout the experiment no nucleation of NaCl was observed in either of the two glass beads layers indicating a nucleation barrier. In the second biotite-bearing sample (SB) we observed indeed the nucleation of an NaCl crystal after the porosity breakdown in the biotite-bearing layer and the loss of connection to the pure NaCl layer. We will explore the origin of this nucleation and possible shift of the Merino effect.

*26) The issue of the glass beads does raise the question of why the authors did not do an additional control compaction experiment with a layer of denser NaCl instead of a layer containing biotite? This would more rigorously test whether the Merino model may apply, i.e. whether diffusive transport is caused by porosity hence supersaturation differences, as opposed to some special effect of biotite. This would be a worthwhile addition to the paper, if time and money allow - as would an experiment substituting calcium fluoride cleavage flakes for biotite flakes. That would be useful because the diffusive properties of NaCl-CaF2 interfaces have been directly measured during active pressure solution of the NaCl by De Meer et al (2002 EPSL 200).*

Thank you for this suggestion. It would have been interesting to explore this aspect as discussed by Schwichtenberg (2021). However, the time frame to run an experiment comparable to the presented experiments would exceed a feasible revision time. Each experiment ideally runs for ~ 2000 hours (~ 3 months) in which the in-house CT scanner has limited availability (e.g. we are restricted to our transmissoin source which precludes large sample scanning). In addition to that, we need to run time consuming image analyses which add a few months to the total time required to get meaningful results.

We agree that there remain interesting questions challenges and which should ideally be addressed in future experiments!

**TECHNICAL ISSUES (language, typographics etc)**

Overall the paper is well written and in good English. Nonetheless a few small improvements can be made as follows:

1. i) an asterisk * is not a mathematical symbol. Proper multiplication and scalar, vector or tensor product symbols should be used.
2. ii) Figures 8-11 would benefit from an explicit indication of which sample is being displayed.

The technical issues will be resolved in the revised manuscript as suggested by the reviewer.

---

## Author Comment (AC2)

**Response to Referee no. 2**

Dear Referee no. 2,

we would like to thank you for your very constructive and detailed assessment of our manuscript. We further appreciate the time and effort that you and referee no. 1 have dedicated to providing your valuable feedback on our manuscript. We are able to include most of your suggested changes which will be highlighted in the revised manuscript.

Please find below our detailed responses to your individual comments.

(*Reviewer comment*; Author's reply)

*The paper by Schwichtenberg et al describes a set of 3 long-term compaction experiments on pure NaCl, a layered sample of pure NaCl and a mixed NaCl/biotite layer, and a layered sample of pure NaCl, mixed NaCl/biotite and pure NaCl. It addresses the question of the role of biotite in pressure solution creep, which is a process relevant to the understanding of deformation processes in the Earth crust. It is not exactly clear how this paper differs in approach and conclusions from earlier work done by Macente et al in 2017 and 2018. The paper concludes that with the type of biotite used, the earlier indicated reinforcing effect of phyllosilicates on pressure solution creep has not been found. The methods and assumptions are valid, and results are probably sufficient to support interpretations and conclusions, provided the two major comments are fixed. Otherwise, the organization of the paper and details of the manuscript are mostly of appropriately high quality, though some edits (see specific and technical comments) are needed to fix what is currently not clear.*

*Apart from the apparent similarity to Macente et al 2017 and 2018, I have two major comments concerning the potential validity of this study.*

*Major comment 1 is related to the technical capacity of the DVC. How well can automatic processing, such as DVC cope, with material literally moving, or jumping, from one place to another? it is written for small amounts of lateral deformation and shape change of particles, so if material moves from one place to another, which the 2D analyses indicate, is DVC then capable of picking it up? The main part of the argument in paragraph 4.3.3 seems to be based on the fact that the code ran and indicated no massive problems, and therefore the answers are correct. This is not necessarily the case. A smaller part of the argument is that the interiors of the grains don't change. But what if new grains are created with a similar shape and size? And what if grains are completely dissolved? In the latter case, a correlation can be made with the neighboring NaCl grain, which looks otherwise quite similar, due to similar initial grain size.*

Major comment 1 contains several arguments to which we will respond separately in the following paragraph:

1. *How well can automatic processing, such as DVC cope, with material literally moving, or jumping, from one place to another? it is written for small amounts of lateral deformation and shape change of particles, so if material moves from one place to another, which the 2D analyses indicate, is DVC then capable of picking it up?*

    The DVC analysis was conducted with SPAM which uses a linear homogeneous transformation function but no higher order shape functions. Hence, it can pick up displacements, rotations, zoom and shearing. The reviewer is correct that the change of shape may cause problems. However, in the present case the dissolution and precipitation process occurs along the grain boundaries while the grain centres are not affected by

deformation. As the texture of the sample is preserved throughout the experiment SPAM correlates very well.

In addition to that, we are looking at very small incremental amounts of deformation as we are comparing two successive time steps with each other. We agree that if we would try to compare the first and last scan of each experiment, we may encounter problems due to too much deformation. We have tried that in the past and it did not correlate well. But for the small deformation steps between successive compaction stages SPAM correlates very well.

2. *A smaller part of the argument is that the interiors of the grains don't change. But what if new grains are created with a similar shape and size? And what if grains are completely dissolved? In the latter case, a correlation can be made with the neighboring NaCl grain, which looks otherwise quite similar, due to similar initial grain size.*

We agree with the reviewer and think this is a valid point. However, we are monitoring the experiments in 4D and any nucleation or complete dissolution of grains would have been visible. In a failed experiment we actually did observe dissolution which was rather easy to spot. New grains on the other hand can only nucleate in the open pore space, therefore will never reach both, same size and shape as the old grains.

As for major comment 1 we will address the individual arguments of major comment 2 separately in the following paragraph:

*The second major comment is related to the starting porosity, a critical element for compaction experiments, and a notoriously difficult one to control. The initial compaction was 9 to 18%, but the starting porosity of the samples is quite different (Figure 12). In the mixed samples this porosity is not homogenously distributed. Since pressure solution is heavily affected by porosity, how does this affect the rates and results you indicate? and on this note, the term steady state compaction is misleading, since the compaction rate should continuously decrease (see references in the manuscript). It is also not entirely clear how porosity is determined: is this like Macente et al from a 400^3 voxel subvolume in the CT scan? If so, include in the method section. Is the determination of the 2D porosity and 2D presence of NaCl per slice, but for the full sample, and for the 3D volumetrics on subvolumes only?*

1. *The second major comment is related to the starting porosity, a critical element for compaction experiments, and a notoriously difficult one to control. The initial compaction was 9 to 18%, but the starting porosity of the samples is quite different (Figure 12). In the mixed samples this porosity is not homogenously distributed. Since pressure solution is heavily affected by porosity, how does this affect the rates and results you indicate?*

A higher initial porosity compared to the pure NaCl sample was observed for the SBS sample (~3%) and could account for higher stresses at grain contacts hence, higher strain rates. However, the initial porosity of the SB sample was lower than the one of the pure NaCl sample, and yet the strain rate was accelerated. We expect other factors such as the effective load to influence the strain rate as well.

2. *and on this note, the term steady state compaction is misleading, since the compaction rate should continuously decrease (see references in the manuscript).*

We agree with the reviewer and will change "steady state" to "apparent steady state" throughout the revised manuscript.

3. *It is also not entirely clear how porosity is determined: is this like Macente et al from a 400^3 voxel subvolume in the CT scan? If so, include in the method section. Is the determination of the 2D porosity and 2D presence of NaCl per slice, but for the full sample, and for the 3D volumetrics on subvolumes only?*

We determined the porosity as 2D porosity per slice of the sample. In case of the salt distribution we used two different approaches. The first one was equivalent to the 2D porosity measurement (per slice of the whole sample) and the second approach determined 3D volumes from subvolumes. We will clarify that in the revised manuscript.

**Specific comments:**

*Line 15: this is the only place where the length scale is actually quantified, whereas it would make sense to include it in the discussion paragraph 4.1.*

Thank you for pointing this out. In the revised manuscript this will be added to the discussion. There, we already mention that the diffusive transport occurs on length scales of multiple grain diameters. A true quantification, however, is difficult as we cannot trace the dissolved material from source to sink, we can only identify the source layer in contrast to the sink layer.

*Line 73: please add a clarification on the different aspect ratio of the biotite flakes. Which dimension is 200-500 microns?*

Line 73 will be updated. The dimension of 200-500 µm is the grain size of the biotite grains, hence the maximum diameter of the grains.

*Line 76: dry NaCl?*

We will clarify this in the revised manuscript. For the pure NaCl experiment we used dry NaCl straight from the container of the chem. compound. In contrast to the NaCl used in the preparation of the layered samples, the "dry NaCl" was not mixed with brine into a slurry.

*Line 80: simple insertion of the piston, or already with a specific applied force?*

As suggested by the reviewer, we will add details about the process in the revised manuscript. The piston was inserted into the sample cell by twisting. The cells were then flushed with pressurized brine. To avoid that the piston was pushed out of the cell by the fluid, a load was applied to the top piston, that kept the piston in place but was low enough so that the effective load on the sample remained zero. The load was calculated for the individual setups of different samples.

*Line 86-91: out of curiosity, why is there a difference between SBS and SB samples in the design of the pumping system? Is there a different brine used? Or is it just one of those things that happens when experiments progress?*

We had to switch to a different pumping system as the glass column of the initial transfer vessel broke during the preparation of the second experimental suite. Further, it was reasonable to operate the system at a lower pore fluid pressure because we aimed to raise the effective stress in individual samples.

*Line 92: what was the fluid pressure? Was this the same for all three experiments?*

The fluid pressure was 5 bar in the SBS experiment and 2 bar in the SB and S2 experiment. In order to make this information accessible in the revised manuscript we will add a summarizing table to the supplementary material.

*Line 98: why is there a difference between the constant effective load for SBS (6.64 MPa) for SB + S1 (10.5 MPa)? What is the load during the experiments? Please add here.*

We will implement the change as suggested by the reviewer. The effective load of each sample remained constant throughout the experiments. The conditions for the first experimental suite (SBS) were chosen to be similar to the ones used by Macente et al. (2018) which allowed comparison of the data with each other. Afterwards the effective load was increased for the second experimental suite (SB and S2) in order to increase the strain rate of the deformation process according to the rate law for diffusion controlled DPC (Spiers et al. 2003).

*Line 142: is for this type of microtomograph the gray scale belonging to 100% NaCl density always the same, regardless of scanning conditions? Because in some CT scanners the grey signal "floats", and in some scanners it is fixed. How is that for this scanner?*

In our system the grey value "floats". To minimize this effect, we chose constant scanning conditions for each scan of a sample. That means we used a constant peak energy, and target power loading as well as exposure time and source-camera distance. Hence all time steps were scanned under constant illumination. Further, we used the same reconstruction parameters for each scan of a series. Hence, the grey signal for an individual phase should in theory be the similar for the scans of a series. Minor differences only affect the segmentation process as classifiers may need to be adjusted between the individual scans of a sample.

*Line 155-157: I do not understand the size of the 3rd dimension for the 3D NaCl subvolume.*

We agree with the reviewer that this needs clarification, and we will do so in the revised manuscript. We picked a biotite grain at the top and one at the bottom of the biotite-bearing layer which were easy to identify in every compaction step. The distance between those two grains defined the 3rd dimension of the NaCl subvolume and decreased with increasing compaction/ progress of deformation.

*Line 176: How do SPAM and TomoWarp deal with grains which change shape themselves? They do not only rotate and rearrange but can also change shape due to dissolution and precipitation (major comment 1).*

SPAM uses a linear homogeneous transformation function but no higher order shape functions. Hence, it can pick up displacements, rotations, zoom and shearing. The reviewer is correct that the change of shape may cause problems. However, in the present case the dissolution and precipitation process occurs along the grain boundaries while the grain centres are not affected by deformation. As the texture of the sample is preserved throughout the experiment SPAM correlates very well.

TomoWarp on the other hand is based on displacements measured by SPAM and therefore not affected by the change of shape as long as SPAM correlates.

*Line 186-187: all samples were under a constant and similar effective vertical load during this compaction time? This doesn't become clear from the preceding sections. What is the starting porosity of the sample? Is it homogeneous throughout the sample? Does each sample have the same starting porosity? (major comment 2)*

We agree with the reviewer and we will include the missing information in the revised manuscript as a summarising table in the supplementary material (vertical loads) and in the results section (starting porosity) of the manuscript.
The effective load was constant for the entire duration of the experiments but varied between the individual samples.
S2 : 10.5 MPa; SB : 10.5 MPa; SBS : 6.64 MPa

The starting bulk porosities (at t=1hr) of the samples are not the same but they are similar to each other.
S2: 27%; SB : 24.3% (Bt), 25.3% (NaCl); SBS : 30.8% (NaCl-top), 30.8% (Bt), 33.0% (NaCl-bottom)
The biotite-bearing layers have a lower initial porosity than the pure NaCl layers which we explain with a higher packing density of bt-grains in combination with NaCl-grains.

*Figure 3 and line 198-206: why the smooth connection between datapoints in Figure 3a? What is the highest resolution in vertical strain rate you can obtain with your measurement method? The fact that a plateau is reached can also mean you have reached the measurement capacity of the setup. In principle, in a pressure solution type of process, based on theory (citations in the manuscript), one would expect a continuously decrease in strain rate with porosity. In other words, it is a steady state in the length of the experiment, but if you could measure indefinitely, the rate would continue to decrease. So is it really a 2 stage process, or is it actually a visual artefact caused by measurement resolution and experiment duration?*

The smooth connection between the data points is a spline interpolation. The reason for choosing an interpolation rather than connecting the data points with each other is that we do not have measurements in between data points. Although we expect the compaction to follow the depicted trend, we cannot exclude positive or negative deviations.

The highest possible resolution of the strain rate is a shortening of one slice over the entire duration of the experiment. That is a strain rate of $8.16e^{-11}$ $s^{-1}$ for SBS, $8.97e^{-11}$ $s^{-1}$ for SB and $1.33e^{-10}$ $s^{-1}$ for S2. Hence, the minimum strain rate reached in the experiments is still orders of magnitudes higher than the resolution of the measurement.

We agree that the term steady state might not be appropriate for the data, a better term would be apparent steady state. We will correct this in the revised manuscript accordingly and replace "steady state" by "apparent steady state".

The two stages of the process are rather related to a transition from a loosely packed aggregate where mechanical compaction significantly contributes to the strain rate, to an interlocked aggregate dominated by chemical compaction.

*Line 225/Figure 7: as Figure 3 and line 198-206: is it caused by steady state or measurement resolution?*

Here, again the maximum resolution of the measurement is defined by a minimum displacement of one slice/pixel per time interval. As discussed for the strain rate in the comment above, the z-displacement rates plotted in Fig. 7 exceed the resolution of the measurement by orders of magnitude.

*Figure 7: this is z-displacement rate. In the NaCl-biotite-NaCl sample both NaCl layers have a different thickness than the mixed layer, where in the NaCl-biotite sample they are of similar thickness. If you would plot strain rate instead of z-displacement rate, would the trend then change?*

Plotting the bulk strain rate gives a similar result as the DVC analysis. Both strain rates are similar to each other.

*Figure 8-9-10: why did you not do the DVC for all time steps? How certain are you that the time steps shown are representative?*

We did the DVC for all time steps and selected the three data sets shown in figures 8-10 after thorough inspection of the results.

*Line 229-245: please be more precise in your description, and in labelling if you are looking at compactive or dilative strain maximum in this paragraph. In Figure 8 (SBS), I see deviatoric strain maxima in the center of the sample, correlating with positive volumetric strain (dilatation), and overall more activity in the bottom half of the sample. In Figure 9 (SB) I see similar high deviatoric strain in the center, but more activity in the top half of the sample. There is barely any dilatation. In Figure 10 (S1), there are high deviatoric strains in the center, and both dilation and compaction, with more activity in the bottom half of the sample. Moreover, what would be the minimum strain needed to be measurable? The samples overall do look blue, but how blue does it need to be to be sufficiently away from zero?*

We will thoroughly revise the section and state more precisely where strain maxima occur within the samples and how they correlate with each other. In addition to that, we will identify the minimum strain for each sample.

*Table 2: in all three figures, there are three plots for the DVC, but only two data entries for each sample in this table.*

As suggested by the reviewer we will add the third value to the table.

*Line 236: I would consider the use of the word "trend" with only two data-points per sample too strong.*

We agree that indeed two values do not define a good trend.

*Line 243: "deviatoric strain maxima corresponded to the location of biotite grains as well as open pore space and pure NaCl clusters" – in other words, there is no correlation between the location of the deviaotric strain maxima?*

Yes, that is correct.

*Line 247: the correlation is not absolute: the maximum loss of porosity in the SB sample (1932 hr) is from slice 500-925 or so, and the biotite layer ends at slice 1000. For the SBS sample, the maximum loss (1932 hr) is from slice 800 to slice 1550, and the biotite layer is from slice 750 to 1350. How does the location of the maxima compare to the data from the DVC?*

The biotite layers have curved boundaries rather than straight ones, so pure NaCl measurements influence the porosity measurements as well. That is the reason why the limits of the maximum porosity loss do not match exactly the boundaries of the biotite-layer. In order to compare the location of porosity maxima to the DVC we suggest to plot the porosity distribution on top of the DVC results and add the figure to the revised manuscript.

*Figure 12: the starting porosity is quite different for the samples. How would this affect the average compaction curves of Figure 3?*

The porosities measured in the very first scan after 1 hour of compaction are ~30% (SBS), ~25% (SB) and ~27% (S2). A higher initial porosity compared to the pure NaCl sample as observed for the NaCl-biotite-NaCl sample (~3%), could account for higher stresses at grain contacts and therefore higher strain rates. However, the initial porosity of the NaCl-biotite sample was lower than the one of the pure NaCl sample, and yet its strain rate was accelerated.

*Line 254-259: how did you determine the NaCl distribution? 100% minus porosity minus biotite? Or did you also segment the NaCl grains individually? What is part of the NaCl remains in solution as supersaturation, as indicated in the discussion as a potential part of the process?*

NaCl was discretely segmented using the Deep Learning tool of Dragonfly as "simple" segmentation by Trainable Weka was not possible.
We can only speculate about the amount of NaCl that remains in solution.
Please see the following calculation for an exemplary estimate:

We had a 5mm OD x 10 mm column of NaCl with 25% porosity, and that porosity was filled with saturated brine at room T then we have:

Volume of column = 196 mm2; Volume of NaCl = 147 mm2; Volume of brine = 49 mm2

Concentration of saturated NaCl brine = 6.15 M

Moles of NaCl in brine = 0.30 millimoles; Moles of solid NaCl = 7 millimoles

% of NaCl in the column present in brine = 4.2%

Desarnaud et al (2014) and Zimmerman et al (2015) indicate maximum supersaturation of 1.6x. Hence, the maximum additional salt in solution through supersaturation would be an additional 2.5% of the total mass of the solid NaCl if all of the solution were at the supersaturation limit for homogenous nucleation.

That calculation sets a maximum upper limit as we're not dealing with homogenous nucleation. The reality would likely be much closer to the limit set by the saturation, and thus well within the likely segmentation error of the NaCl segmentation.

*Line 260-264: Unclear phrasing: if the assumption is made that biotite is an insoluble internal standard (line 261), it makes sense that the analyses show the biotite content to be standard… And can you show somewhere in a Figure where the subvolume is taken (this would also solve line 155-157)?*

We assume that the reviewer means "constant" instead of "standard". Line 261 will be rephrased accordingly. In addition, we will add a map for orientation in the supplementary material of the revised manuscript.

*Line 273-275: it is not clear to me why this is interpreted a change in deformation mode, instead of it being a continuous log-linear decrease in rate (same comment as in the description of the results).*

Please see response to the comment concerning figure 3. We agree that this should not be interpreted as a change in deformation mode and will adjust the revised manuscript accordingly.

*Line 278: This needs more careful phrasing, since even the current description of results indicates that strain maxima occurred mainly within the biotite part of the sample (line 233).*

In line 232-233 we write "strain maxima in the biotite-bearing samples were located within the biotite-bearing layer, but not exclusively. Pure NaCl domains were also affected by high

strains…" While we appreciate the reviewer's feedback, we respectfully disagree. We do not think it gives the impression that strain maxima were mainly located in the biotite-bearing layer. In the revised manuscript however, we will rephrase it so that it becomes even clearer that strain maxima occurred in the biotite-bearing domain as well as in the pure NaCl domains. Line 278 will be adjusted accordingly.

*Line 294: unless one takes it that the patterns of Fig 8, 9 and 10 do show there is more strain localization in the biotite… Or that the DVC actually doesn't cope very well with the material transport (major comment 1).*

Thank you for pointing this out. We agree that locally higher strains might occur in the biotite-bearing layer. However, the bulk magnitude of strain is not higher. Further, the correlation fields in the supplementary material show that DVC correlates very well even in chemically changing samples.

*Line 329: This wasn't clear to me in the results on the DVC, though the concentration of deformation was mentioned in Figure 12 and 13. Perhaps it would help to add arrows or boundaries to Figs 8-9?*

We agree that Figs. 8 -9 do not display the location of shear strain on the grain scale clearly enough. In order to better visualize the correlation, we will implement the reviewer's suggestion and highlight biotite-NaCl phase boundaries in the figures.

*Line 333: I do not understand how figure 5 demonstrates the efficiency of this process*

In figure 5 you can see a single NaCl grain which is in contact with two biotite grains. With progressing deformation, you can see that the NaCl grain is reduced in size without showing any signs of brittle deformation. We interpret this to happen due to pressure solution along the interphase boundaries between the NaCl and biotite grains. We will adjust the figure caption in the revised manuscript to clarify the case.

*Line 334: ah, that's what the Lambert plots did (technical comment line 180)! But if there is no significant rotation, then why is the deviatoric strain so high in the biotite layers? Another reason could be that many of them are already fairly horizontal, so that might also be why there is no strong realignment.*

We agree with the reviewer that right from the beginning many biotite-grains are already horizontally aligned, however if a biotite grain is point loaded on one side and is not subject to effective dissolution one would expect that the mechanical compaction of the aggregate causes rotation of the biotite.
The deviatoric strain can result from e.g. grain boundary sliding during DPC. Rotation is not the only source that can cause deviatoric strain. Also, grain boundary sliding should cease in a denser aggregate, which we observe as deviatoric strain rates are decreasing over time.

*Line 345: can you add here that Macente reported a first order effect (i.e. why would you expect a first order effect), and which observations showed there is no first order effect?*

> Unfortunately, Macente (2017) and Macente et al. (2018) did not analyse the biotite composition in their study and therefore, we cannot add as requested by the reviewer, that they reported a first order effect.
> The analysis of the chemical composition of the two types of biotite shows that although the compositions are different, our results are comparable to Macente et al. (2018). Both studies show a similar effect of biotite upon the porosity. In theory a different chemical composition can affect the dissolution process by either enhancing or impeding it. This was not investigated in our study, and consequently no effect could be observed.

*Line 367: why/how does Figure 11 show that local maxima correspond to sites of precipitation?*

> The volumetric strain maxima correspond to the NaCl-pore interface which is located in between e.g. the biotite grains. We interpret the NaCl-pore interface as active site of precipitation. We suggest the addition of a plot of the porosity on top of the volumetric strain pattern to clarify the correlation.

**Technical comments**

*Line 62: "which are described in Macente (2017)": Since the description is actually below, this phrasing is slightly misleading*

> We agree with the reviewer. Line 62 will be rephrased to "…oedometer cells (Fig. 1). A detailed description of the cell design can be found in Macente 2017."

*Line 105: for clarity, it would be nice to add if the samples were compacting in the same building (I assume so), or if they were transported by car throughout Edinburgh or the UK or even from France (looking at the affiliations of the authors). Given the composition of the author team I imagine the transport between CT scans and compaction location was done carefully, but the explicit mention of the location of the tomography instrument somehow gives the impression that the scans were done somewhere far, far away… Which would have consequences for their validity.*

> We agree that this has a potential effect on the study. Significant transportation of the cells for each scan would have had consequences for the study, which is why we didn't do the experiments and scans in separate locations.

*Line 106-107: how many scans and compaction time for the S1 sample?*

> Thank you for pointing this out. We will add the parameters for the pure NaCl sample in the revised manuscript. We took a total of 5 scans over a duration of 1089 hours.

*Section 2.5: this section would be easier to read if there was a flow diagram that briefly labels all the steps and different softwares*

> We think this is an excellent suggestion. We will add a flow diagram in the revised manuscript.

*Line 136: please mention your figures in order of appearance. Fig 12 now follows Fig 2. Fig. 12 doesn't contain the error, though that is suggested by this part of the text. Idem for Fig 13 and Fig 14*

> We appreciate the reviewer's feedback on the order of figures. We respectfully disagree with a change of this order as the text in line 136 (methods section) refers to a figure in the result section in order to give a visual example of the plotting method. Hence, we cannot avoid that Fig. 12 follows Fig. 2.
>
> We also agree that according to the text errors should be plotted in Figs. 12, 13 and 14. We will add these in the revised manuscript.

*Line 159: given the name (digital \*volume\* correlation) I assume this approach is only valid for the 3D volumes, correct? Please add.*

> The reviewer is correct, DVC is valid for 3D volumes in contrast to DIC for the 2D case. We will add this accordingly.

*Line 160: can you indicate in 1-2 lines which operations or calculations are performed by SPAM and which by TomoWarp2?*

> We agree with the reviewer that this would help the accessibility of the method therefore, we will add this to the revised manuscript.
> SPAM calculates the displacement field, while TomoWarp uses the displacement filed as an input to calculate the strain field.

*Line 180: this is my own ignorance: how does one read a Lambert projection? As the reader, what would it tell me? Can you add a reference here so the non-knowledgeable reader can read up on the importance of these plots?*

> Thank you for pointing this out. A Lambert projection is a conic map projection. Here, the projection sets a cone over a sphere and projects the surface conformally onto the cone. The cone is unrolled, and the parallel that was touching the sphere is assigned unit scale. (From Wikipedia)
>
> In our case the parallel that was touching the sphere represents the equator of the sphere. Similar to a stereonet, the Lambert projections used in this paper show the upper half of a sphere. The center point of the Lambert projection is the pole of our sphere.
> Similar to a stereonet, the outline of the Lambert projection represents the orientation on a plane ("dip direction") while the circles represent the "dip".

Reading the Lambert projection is similar to reading a stereonet for poles.
We agree that a reference should be added to the revised manuscript.

*Figures 4 and 5: why did you choose this specific vertical slice? Where is it located in the 3D sample? Would we see the same if you choose any other slice?*

We chose the middle slice of each sample because it was the easiest to identify in every compaction step. However, other vertical slices show similar patterns and could have been used equally.
We suggest adding a sketch of the location of the slice in the sample to fig. 4 as it is the same for all three samples.

*Figure 5: can you add the red markers to all 5 panels? It would help guide the eye. The lower biotite grain seems to also change curvature between the panels, or is that simply due to the unfocused visualization?*

We appreciate the reviewer's feedback and will add them to all grains. The curvature of the biotite grain can be either the result of unfocused visualization or a cutting effect, after all the system is reacting and we cannot guarantee 100% to always find the exact same slice through the grain.

*Figure 5: Why do you not have panels also to show if similar things happen in the SB and SI samples?*

The same effect can be observed in the SB sample and we can add such a panel to the revised manuscript and also one of the pure NaCl sample for comparison as the process is less pronounced there. We decided against doing so in the original manuscript as it would not add indispensable information to the manuscript.

*Figure 6: to my non-Lambert-trained eye, figures a and b look very similar… Why could you measure so much more grains for a versus b? Is that because there were more grains in b to keep the layers of equal thickness?*

We agree with the reviewer, the figures are indeed very similar. Which shows that biotite grains in both samples are already horizontally aligned at early stages of the experiments and do not rotate much.
The reason why we were able to measure more grains in b than in a is that we always included 20wt% biotite in our biotite-bearing layer. The layer is thicker in the SB than in the SBS sample hence, it contains more biotite grains.

*Line 229-234: For readability, please treat the descriptions in the same order as the figures are shown for clarity, and in Figure 8 9 and 10 please add the sample name in the caption or in the figure. This could be improved throughout the paper: sometimes the pure salt sample is described first, and sometimes the salt-biotite-salt sample.*

We agree that the reviewer's suggestion can improve the readability of the manuscript. We will rearrange figures and text passages to a consistent order which will be used throughout the revised manuscript.

*Figure 8 9 10: compaction in rock mechanics experiments is often denoted positive, whereas here the negative values are compaction (line 234/second-last line of caption).*

We appreciate the reviewer's assessment. The reason for this switched notation is that SPAM has its origin in engineering rather than rock mechanics. We agree it can be confusing when coming from a rock mechanics background, however, to be able to compare published DVC results with each other which to our knowledge all use the engineering definition, we would like to keep compaction negative.

*Figure 9: typo: "cumulative"*

We will change that.

*Paragraph 4.1: the title of the paragraph, combined with the question of the introduction, gives the reader the impression the length scale will be quantified, whereas this is actually a more qualitative interpretation.*

We will rephrase the title of the subsection 4.1 in order to clarify that the length scales of diffusive transport during DPC are not quantified in this manuscript.

*Line 374 – 388: OK, but how can you then be sure for the rest of your sample that the values are correct? You probably can I'm sure, but I don't see it straight away. What am I missing?*

After thorough revision of the section we decided to remove it in the revised manuscript. The presented mathematical derivation of the isotropic strain is based on the small strain theory while we used the large strain theory for the strain determination in our DVC analysis. While it does not explain the positive volumetric strain in the glass beads layers (SBS sample, first increment), it does however explain why we chose large strain over small strain as the basis for our strain measurements.

*Given the length of appendix A2 and how crucial the terms are, I suggest to move this definition to the method section.*

We will include the definition of strain as used throughout the manuscript in the methods section of the revised manuscript.

---

## Author Response (AR2)

Dear Editor,

we would like to thank you and the two referees for a very accurate and constructive revision of our manuscript. We appreciate the time and effort that you and the referees have dedicated to providing your valuable feedback on our manuscript. We have carefully considered the referee's comments and tried our best to address each one of them to improve our manuscript. The changes will be highlighted in red and blue in the revised manuscript. Please find below our detailed point-to-point responses to the individual comments.

**Referee's comment:**

1) *Section 4.1., para 2. The authors state that one reason that the DVC method works is because grain centres did not deform. Can this be explained somewhere, e.g. in the method. Or perhaps give a reference that explains this. Would the method really not work if the aggregates had been deformed by entirely crystal plastic deformation? I ask this because, while the authors claim that plastic deformation would have been restricted to small grain contacts, there is plenty of literature showing significant crystal plastic strains in NaCl single crystals at differential stresses in the range 5-10 MPa at room T.*

**Authors' response:**

We have addressed the referee's concern in Section 4.1, lines 303-314, where we clarify why DVC can be used in our case as a criterion for the negligible effect of crystal plastic deformation upon the bulk deformation of our samples.

**Referee's comment:**

2) *Fig 16. The authors have made a serious attempt to address the issue of consistency between their compaction data and previous data in this figure. However, they have compared rates at similar times since the onset of compaction, rather than making the comparison at similar values of porosity, which is the relevant microstructural state variable here (see the cited papers on the porosity/strain dependence of compaction rate by p-sol). Comparison at similar times really has no physical meaning. This may be why the authors obtain up to 2 orders discrepancy. I would urge a comparison at similar porosity, or else avoid any claim that the compaction data are "in accordance" with previous data beyond pointing out the qualitative similarity with previous compaction curves, except for the attainment of an apparent steady state. Previous compaction data on analytical salt and salt backfill materials (e.g. WIPP site) consistently show continuously decelerating creep, even over periods of years.*

**Authors' response:**

We agree with the referee and have changed Fig. 16 accordingly. In its updated form it compares the strain rate at similar porosity not compaction time. In the accompanying text, we further avoided to claim consistency with previously published data beyond a qualitative similarity.

We hope that you feel the revised manuscript adequately addresses the referee's concerns.

Best wishes,

Berit Schwichtenberg